# What's Different about Bank Holding Companies?

**Ralph Chami** [1], **Thomas F. Cosimano** [2,*] , **Jun Ma** [3] **and Celine Rochon** [1]

1 Institute for Capacity Development, International Monetary Fund, 700 19th Street NW,
Washington, DC 20431, USA; rchami@imf.org (R.C.); crochon@imf.org (C.R.)
2 Department of Finance, Mendoza College of Business, University of Notre Dame, Notre Dame, IN 46556, USA
3 Department of Economics, Northeastern University, Boston, MA 02115, USA; Jun.Ma@Northeastern.edu
* Correspondence: t.f.cosimano@gmail.com

**Abstract:** We develop a dynamic model of a BHC that encompasses both a trading desk and a loan desk, and explore the role of risk attitude and overleveraging by the trading desk. We trace the impact of monetary policy and market innovations on bank behavior in the presence of Basel III type regulations. We show that the value of the BHC is enhanced by operating both desks. We explore alternative regulatory remedies to ongoing efforts to ring-fence the proprietary trading business, and show that regulations that target bank governance can mitigate possible rogue trading and the overleveraging problem.

**Keywords:** bank holding company; term structure; financial markets; banking; Basel III; bank capital; financial stability; monetary policy; macro prudential; ring-fencing

**JEL Classification:** E43; E44; E5; G2

## 1. Introduction

Understanding the link between monetary policy and financial stability hinges on identifying the channels of transmission. As efforts to deal with the financial and economic fallout from the great recession clearly demonstrate, banks, and particularly bank holding companies (BHCs) with trading desks, play a key role in propagating monetary and financial shocks to the rest of the economy. In addition, post-crisis regulatory frenzy targeting BHCs is influencing their behavior and profitability, as well as the rest of the economy. For example, efforts by US policymakers and bank regulators to rein in trading operations of BHCs have resulted in a number of US based BHCs shedding off their proprietary trading while others have moved their operations overseas.[1] More generally, policy makers-especially in the aftermath of the great recession-have been interested in understanding the reaction of systemically important financial intermediaries (SIFIs) to the monetary policy stance, and to changes in micro and macro prudential policies aimed at enhancing financial stability. In this paper, we analyze the behavior of BHCs and their interaction with financial markets when subject to regulatory constraints. In particular, we focus on their trading and lending operations in reaction to market and monetary shocks and to changes in Basel regulations, and highlight the resulting implications for credit supply.[2]

SIFIs include large commercial banks, and financial as well as bank holding companies (FHCs and BHCs), among others.[3] Data indicate that BHCs represent 20.51% of domestic financial sectors assets and 99.12% of US GDP (using second quarter 2015 for the US), with the top 5 BHCs accounting for 51% of total BHC assets.[4] These large intermediaries typically encompass both lending and securities trading operations, see Table 1. Generally, there's an active trading desk whose book is marked to market, and that is tasked with managing bank liquidity and interest rate risk. This trading desk exists alongside a lending operation, where loans are priced at book value. While the trading operation offers ways

for the bank to manage interest rate risk it also could be a source of risk for the rest of the bank.

**Table 1.** Financial Ratios for the Top 5 BHCs.

| Date | Stat | Tier 1 | Leverage | Loans | Securities | Deposits |
|------|------|--------|----------|-------|------------|----------|
| March | Mean | 10.28 | 11.98 | 44.80 | 22.93 | 70.08 |
| 2016 | Std | 3.42 | 2.74 | 8.70 | 16.25 | 5.23 |
| March | Mean | 6.72 | 8.92 | 42.44 | 24.24 | 54.74 |
| 2008 | Std | 0.78 | 1.03 | 20.40 | 5.57 | 17.92 |

We show that such a BHC behaves quite differently from a bank with only a lending operation. In particular, the trading desk confers benefits to the bank through its management of interest rate risk. This benefit arises because the trading desk has the flexibility to take either short or long positions in treasury securities while the loan desk is charged with issuing illiquid longer term loans. This benefit results in higher capital and profitability to the bank. On the other hand, it can also impose additional risks on the lending operation and on the overall bank due to overleveraging, risky trading behavior, or simply due to wrong bets or expectations regarding future yield rates.

Given their size and impact on local and global markets, researchers, policymakers, and regulatory bodies have been busy trying to understand how best to regulate their behavior. Yet the academic banking literature, for the most part, has mainly focused on models of banking with lending operations that typically transform liquid deposits into longer-term (illiquid) loans. These models have constituted the main framework for understanding bank behavior and their reactions to monetary and regulatory policies, and for informing the discussions on optimal regulatory policies. (See Bernanke (1983); Diamond and Dybvig (1983); Bernanke et al. (1996, 1999); Bernanke and Lown (1991); Bernanke and Blinder (1988); Berrospide and Edge (2010); Calomiris and Mason (2003); Hancock and Wilcox (1993, 1994); Kiyotaki and Moore (1997); and Peek and Rosengren (1996)). Of course, this is understandable, as, traditionally, banking crises have been credit and liquidity risk crises. The 2007 financial debacle, however, highlighted the role of capital markets and market risk along with credit risk in initiating and propagating the crisis. See Adrian and Shin (2010); Kashyap et al. (2014); and Brunnermeier (2009). These sources of risk coexist on the same balance sheet of a BHC, and could potentially interact and affect the overall risk profile of the bank with implications for financial stability and the rest of the economy.

The question that arises is whether the conclusions regarding bank behavior and regulatory policy prescriptions gleaned from using only the lending side of the banking business would continue to hold when the trading business is operating alongside the lending business. In this paper, we develop a dynamic model of a BHC that encompasses both a trading desk and a loan desk. We study the behavior of such a bank and the impact of monetary policy innovations on BHC behavior in the presence of Basel III type regulations. To our knowledge, this is a first such exercise.[5]

The BHC, in this paper, operates in an oligopolistic market and maximizes the present value of all future profits under capital and liquidity constraints a la Basel III.[6] In Section 2 financial markets are represented by a continuous time affine term structure model of yield to maturity.[7] This is summarized by three yield curve factors that represent the level, slope and curvature of the yield curve, and can be interpreted as providing information about inflation, the business cycle, and financial crisis. We trace the impact of shocks to the term structure on the hedging behavior of the trading desk, loan pricing decisions, balance sheet composition, capital allocation within the two business lines, and on credit provision. Section 3 introduces the BHC framework, presents some stylized facts about these banks, and lays out the capital and liquidity constraints associated with Basel II and III, including liquidity coverage ratio (LCR), net stable funding ratio (NSFR), as well

as the counter cyclical buffer requirements. Section 4 introduces the optimal portfolio of treasury securities which is chosen by the trading desk. These portfolios allow for both long and short positions in various maturities of treasury securities based on the expected future path of the yield curve factors.[8] Given our estimated term structure, we show in an experiment that the portfolio is long longer term treasury securities and short shorter term securities, when the yield curve factors are above their long term values. On the other hand, the portfolio is short longer term securities, when the factors are below the long term factors. As a result, the trading desk has expected capital gains when the yield curve factors are closer to their long term mean and expected capital losses for extreme increases or decreases in these factors. We conclude this section with an exploration of leverage and risky trading behavior by the desk manager.

To separate out the decisions made by the bank a chief operating officer (COO) is introduced in Section 5. This COO decides the scale of the trading and lending operations by allocating the current capital of the bank between these two operations based on the expected marginal value of capital for each operation. The COO also considers whether or not to raise additional capital in the next period by issuing more capital or changing dividends paid to the shareholders of the bank.

The COO's decisions are based on the expected marginal value of the bank's capital. This expected marginal value is based on the optimal decisions of the trading and loan desks conditional on the current values of the yield curve factors. The trading desk's expected marginal value of capital is dependent on the current values of the yield curve through the expected gross growth rate of capital under trading desk's optimal portfolio. The expected marginal value of capital for the loan desk is dependent on the use of the bank's capital to satisfy the Basel III constraints. The loan decisions follow traditional banking models in that the loan desk takes liquid deposits as given and decides how much illiquid loans should be issued by comparing the marginal revenue with marginal cost of loans. Under normal circumstances the marginal revenue of loans is equal to its marginal cost, yet bank capital has positive marginal value, when the Basel III constraints are binding. In these cases the shadow prices for these constraints are positive, so that there is an additional component of marginal cost. We find that there is a critical level of the loan rate such that the Basel III constraints exactly bind. If the shocks to the demand for loans and the yield curve factors lead to a higher loan rate desired by the bank, then the Basel III constraints are binding and the bank's capital is valuable. This means that the expected marginal value of capital is an option which pays off when the loan rate is above its critical level. We prove that this option is a long straddle, which pays off when the yield curve factors are extreme values, which occurs when the trading desk has a significant capital loss. Thus, the capital of the bank is an insurance purchased and allocated by the COO to insure the loan desk against binding Basel III constraints, when the yield curve factors are closer to their long term mean. However, there are extreme values of the yield curve factors in which the insurance contract fails, the Basel III constraints becoming binding, and the probability of bank distress increases.

In the final section, we summarize the implications of the analysis for the management and regulation of BHC following Sections 6–8 of Chami et al. (2017)

## 2. The Financial Market

The Treasury yield to maturity, $r_{\tau,s}(X(s))$, is driven by an affine process, relating this yield of each maturity to the $N$ underlying factors, $X(s)$, such that:

$$r_{\tau,s}(X(s)) = A_\tau + B_\tau X(s). \tag{1}$$

The time subscript $s$ corresponds to today's date, and $\tau$ is the maturity date. The parameters $A_\tau$ and $B_\tau$ for each maturity are set so that there is no arbitrage opportunity for investors in the financial markets.

These yields to maturity will be related to the risk free rate over the term to maturity for the various bonds. It is assumed that the risk free interest rate $r(s)$ is also a linear function of the interest rate factors:

$$r(s) \equiv r(X(s)) = \delta_0 + \delta_1 X(s). \tag{2}$$

The constant $\delta_0$ and the vector $\delta_1$ are independent of time.

The dynamics of the mean reverting stochastic process describing the factors, $X(s)$, under the actual probability distribution, is

$$dX(s) = \left(\gamma^{\mathcal{P}} - A^{\mathcal{P}} X(s)\right) ds + \Sigma_X d\epsilon_s. \tag{3}$$

$\epsilon_s$ is a Brownian motion which characterizes the uncertainty in the interest rate factors $X(s)$. The vector $\gamma^{\mathcal{P}}$ and the matrix $A^{\mathcal{P}}$ are constants, which determine the stationary mean of the factors, $\left(A^{\mathcal{P}}\right)^{-1} \gamma^{\mathcal{P}}$, and the half life of shocks to the factors. The matrix $\Sigma_X \Sigma_X'$ is the variance-covariance matrix for the shocks, $d\epsilon_s$, to the factors.

The solution of (3) for the interest rate factors at the next period relative to its stationary value, $\bar{X}$, is

$$X(t+\tau) - \bar{X} = e^{-A^{\mathcal{P}} \tau}(X - \bar{X}) + Y_\tau, \tag{4}$$

where

$$Y_\tau = \int_0^\tau e^{-A^{\mathcal{P}}(\tau-s)} \Sigma_X d\epsilon_s. \tag{5}$$

The first term in (4) is the percentage of the deviation of the current interest rate factors, $X$, from its stationary value that persists until the next period. The second term is the random changes of the shocks to the interest rate factor from time $t$ to $t + \tau$. This random shock has a normal probability distribution with mean 0 and variance covariance matrix $\sigma_Y(\tau)$.[9]

To carry out risk neutral pricing of zero coupon bonds of various maturities, the actual distribution of the factors is changed through a change of variable which accounts for the price of risk. As a result, the dynamics of the process for the factors, $X(s)$, under the risk neutral distribution, is

$$dX(s) = \left(\gamma^{\mathcal{Q}} - A^{\mathcal{Q}} X(s)\right) ds + \Sigma_X d\epsilon_s^{\mathcal{Q}}. \tag{6}$$

The vector $\gamma^{\mathcal{Q}}$ and the matrix $A^{\mathcal{Q}}$ are the risk adjusted parameters for this process in which the variance-covariance matrix remains the same, $\Sigma_X \Sigma_X'$.

The price of risk in the financial markets is assumed to be affine in the underlying factors.

$$\Lambda(X(s)) = \lambda_0 + \lambda_1 X(s), \tag{7}$$

so that the change of variable from the physical to the risk neutral distribution is

$$\gamma^{\mathcal{Q}} = \gamma^{\mathcal{P}} - \Sigma_X \lambda_0 \text{ and } A^{\mathcal{Q}} = A^{\mathcal{P}} + \Sigma_X \lambda_1. \tag{8}$$

The expected stochastic discount factor conditional on information at time $t$, i.e., $X_t = X$ is given by[10]

$$E_t\left(\frac{M_{\tau,t}}{M_{t,t}}\right) \equiv \mathcal{M}(\tau, X) = \mathcal{M}(\tau) \exp\left\{ -\frac{1}{2}\left(X - \mu_{\mathcal{M}}(\tau)\right)' (\sigma_{\mathcal{M}}(\tau))^{-1} \left(X - \mu_{\mathcal{M}}(\tau)\right) \right\}, \text{[11]} \tag{9}$$

and its random component at time $t$ from state $X$ to $Y$ at time $t + \tau$ is[12]

$$p_M(t, X, \tau, Y) = \frac{\exp\left\{ -\frac{1}{2} Y'(\sigma_M(\tau))^{-1} Y \right\}}{\sqrt{(2\pi)^N \sigma_M(\tau)}}, \text{ such that } \frac{M_{\tau,t}}{M_{t,t}} = \mathcal{M}(\tau, X) p_M(t, X, \tau, Y). \tag{10}$$

Finally, the zero coupon bond price is determined by the expected risk free bond over the maturity of the bond under the risk neutral distribution for the factors.

$$P_{\tau,s} \equiv \exp[-r_{\tau,s}\tau] = E_s^{\mathcal{Q}} \exp\left[-\int_s^{s+\tau} r(u)du\right] = \exp[a_\tau + b_\tau \cdot X(s)]. \tag{11}$$

The first equality follows from the expectation being calculated under the risk neutral distribution conditional on the information at the current time $s$, $E_s^{\mathcal{Q}}$. The second equality follows from the no-arbitrage assumption used to calculate the coefficients $a_\tau = -\tau A_\tau$ and $b_\tau = -\tau B_\tau$ for the bond which matures at $s + \tau$. These coefficients satisfy differential equations which ensure the expected instantaneous holding period return for maturity $\tau$ is equal to the risk free rate over the same period.

Here, the holding period return is given by

$$\begin{aligned}\frac{dP_{\tau,s}}{P_{\tau,s}} &= [b_\tau \Sigma_X \Lambda(X(s)) + r(s)]ds + b_\tau \Sigma_X d\epsilon_s \\ &= \left[b_\tau\left((\gamma^{\mathcal{P}} - \gamma^{\mathcal{Q}}) - (A^{\mathcal{P}} - A^{\mathcal{Q}})X(s)\right) + r(s)\right]ds + b_\tau \Sigma_X d\epsilon_s.\end{aligned} \tag{12}$$

Thus, the expected excess return for a zero coupon bond of maturity $\tau$, $E_t\left(\frac{dP_{\tau,s}}{P_{\tau,s}}\right) - r(s)ds$, is the product of its price elasticity, $b_\tau$ with respect to the interest rate factors and the price of risk for all financial instruments $((\gamma^{\mathcal{P}} - \gamma^{\mathcal{Q}}) - (A^{\mathcal{P}} - A^{\mathcal{Q}})X(s))ds$.

*Estimates of the Term Structure*

We estimate the above term structure model using the monthly unsmoothed Fama–Bliss US Treasury yields data.[13] To keep our analysis within as homogeneous a monetary policy regime as possible and at the same time to avoid the regime of zero lower bound following the most recent financial crisis, we use the sample period from 1999M01 to 2007M12. In our estimation, we use 12 maturities of 3 and 6 months, 1, 2, 3, 4, 5, 6, 7, 8, 9, and 10 years. The continuous time model is estimated using discrete data following the procedure as discussed in Harvey (1990). In this procedure, the continuous time processes (3) and (6) are integrated over a month which leads to

$$X_t = \Theta + \Phi X_{t-1} + \Sigma \zeta_t. \tag{13}$$

Here, the mapping between the continuous and discrete time processes under the physical distribution is:

$$\Theta = \left[I - e^{-A^{\mathcal{P}}}\right]\left(A^{\mathcal{P}}\right)^{-1}\gamma^{\mathcal{P}}, \quad \Phi = e^{-A^{\mathcal{P}}}, \quad \Sigma\zeta_t = \int_0^1 e^{-A^{\mathcal{P}}s}\Sigma_X d\epsilon_s,$$

$$\text{and} \quad \Sigma\Sigma = \int_0^1 e^{-A^{\mathcal{P}}s}\Sigma_X\Sigma_X' e^{-A^{\mathcal{P}}s}ds. \tag{14}$$

Once the continuous time model is transformed to discrete time we use the Kalman filter of the state space model with latent factors explained by (13). The observation equation is given by the yield to maturity (1) plus a measurement error, $\eta_t$

$$r_{\tau,t}(X(t)) = A_\tau + B_\tau \cdot X(t) + \eta_t. \tag{15}$$

This state space model is estimated subject to the no arbitrage conditions, which determine the coefficients $A_\tau$ and $B_\tau$, along with the mapping from the discrete time to continuous time parameters (14). The Kalman filter yields the conditional normal distribution for the factors with conditional mean and variance covariance given by

$$X_{t|t} \equiv E[X_t|r_{\tau,t}] \text{ and } P_{t|t} \equiv E\left[(X_t - X_{t|t})'(X_t - X_{t|t})|r_{\tau,t}\right]. \tag{16}$$

Consequently, the bank has an optimal forecast of the holding period return given by

$$E\left[\frac{dP_{\tau,s}}{P_{\tau,s}}|r_{\tau,t}\right] = \left[b'_\tau\left((\gamma^{\mathcal{P}} - \gamma^{\mathcal{Q}}) - (A^{\mathcal{P}} - A^{\mathcal{Q}})X_{s|t}\right) + \delta_0 + \delta_1 X_{s|t}\right]ds.$$

We use three factors in the state equation since three principle components explain 99.92% of the cross section variations of the 12 yields to maturities. Typically, the three factors are referred to as the level, slope and curvature factors. It turns out that the estimated three latent factors from our model are closely related with these three factors. Following Diebold and Li (2006), we define the empirical level factor to be *yields* (10 *years*), the empirical slope factor to be *yields* (10 *years*) − *yields* (3 *months*), and the empirical curvature factor to be 2 ∗ *yields* (2 *years*) − *yields* (3 *months*) − *yields* (10 *years*). Figure 1 plots the estimated latent factors from the term structure model together with the empirically constructed three factors described above, after proper standardization.[14] These graphs show that the estimated latent factors from the term structure model well track the empirical level, slope, and curvature factors.[15] The plot of level factors reveals that the yields curve level overall has declined throughout the whole sample period. It is important to notice how well the model performs in terms of approximating all three factors. Specifically, the second latent factor and the empirical slope factor both declined for a number of years since mid-2003 until mid-2005 when they started to go up till the end of 2007. However, both the third latent factor and the curvature factor have decreased since mid-2005.

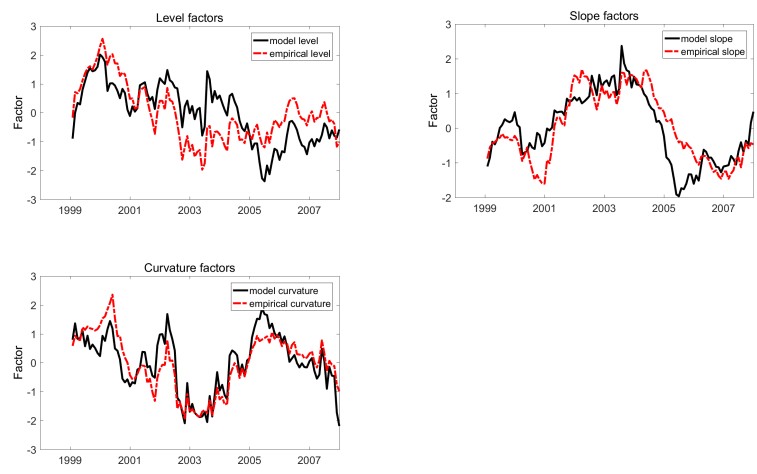

**Figure 1.** Estimated Factors versus Empirical Factors Explaining Yields to Maturity.

Given the estimate of the parameters for the yield curve we can examine the properties of the stochastic discount factor implied by the yield curve in the U.S. from 1999–2008. The expected (deterministic) (11) and random (10) components of the stochastic discount factor are graphed in Figure 2 using the parameters in Table 2. In Figure 2 (left hand graph (LHG)), at the stationary level of the yield curve, $\bar{X}_1 = -0.0177$, the conditional expected stochastic discount factor is 0.8812. Its maximum is 0.9879, which occurs at $X = \mu_{\mathcal{M}}(\tau) = -0.0228$. Thus, there can be a 12% expected fall in the market's valuation of securities for a 0.5% increase in the level. In general, the impact of an expected decrease in the level of the yield curve is positive for $X > \mu_{\mathcal{M}}(\tau)$ and negative for $X < \mu_{\mathcal{M}}(\tau)$. Finally, the standard deviation of the expected stochastic discount factor is more than four times that of the random component of the stochastic discount factor.

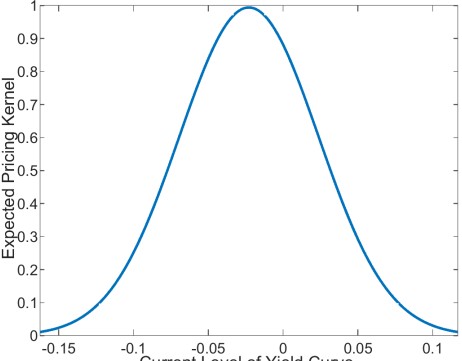
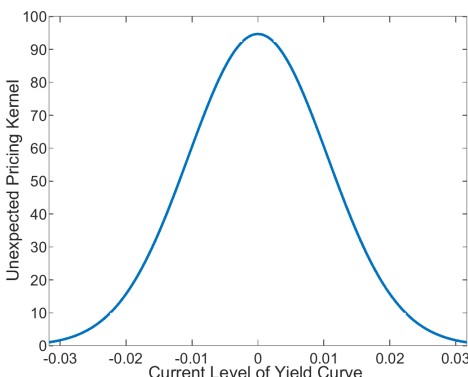

**Figure 2.** Expected (9) and Random Components of the SDF (10).

**Table 2.** Mean and Standard Deviation for Expected and Random Components of the SDF.

| $\mathcal{M}(\tau)$ | $\mu_{\mathcal{M}}(\tau)$ | $\sigma_{\mathcal{M}}(\tau)$ | $\sigma_M(\tau)$ |
|---|---|---|---|
| 0.9940 | −0.0228 | 0.0465 | 0.0106 |

## 3. The Bank Holding Company Model

We define a bank holding company (BHC) as a financial institution which undertakes both trading and lending activities, and is subject to Basel type regulations. A chief operation officer (COO), at the beginning of each period, allocates capital to the two business lines, decides on trading desk leverage, loan rates, and whether to pay dividends or to issue new equity.

The trading desk invests the capital allocated by the COO in marketable securities, which are mainly US Treasuries. To trace the role of the trading desk's attitude toward risk in affecting the trading strategies and the overall bank, we endow the trading desk manager with a constant relative risk aversion utility function and a given leverage ratio. The trading desk manager maximizes the present value of his utility by choosing how much to invest in various maturities of the marketable securities.

The problem is solved in three steps. First, the trading desk chooses the optimal combination of marketable securities, given capital allocated by the COO. The marketable securities are marked to market and are continuously evaluated using an affine term structure model (a continuous time version of Joslin et al. (2011)). Next, given the solution to the trading desk's problem, the COO, who manages a portfolio of loans of various maturities subject to Basel III regulation, sets the interest rate margin relative to the yield on the treasury security with the same maturity. In the final stage, the COO decides on the optimal allocation of capital across the two business lines.[16]

### 3.1. Regulatory Constraints

The balance sheet of bank $j$ consists of reserves $R^j$, two maturities of loans, $L^j$, and treasury securities, $T^j$, for four maturities on the asset side of the bank's balance sheet. Theses assets are funded by deposits, $D^j$, and capital, $K^j$, on the liability plus net worth side. The bank chooses loans and the total capital stock at discrete intervals $t = i\tau$ for $i = 1, 2, \cdots$. All the other assets and liabilities are allowed to change at any time. Consequently, the bank's balance sheet at time $s$ is given by:

$$R_s^j + L_{\tau,s}^j + L_{2\tau,s}^j + L_{2\tau,s-\tau}^j + \sum_{i=1}^{4} T_{i\tau,s} P_{i\tau,s} = D_s^j + K_s^j, \tag{17}$$

for $s \in [t, t+\tau]$, where $\tau$ is the time to the next loan portfolio decision. The treasuries are of four types, allowing the bank to continuously trade over the interval $[t, t+\tau]$: short term

treasuries issued at $t$ and maturing at $t + \tau$, $T^j_{\tau,s}$, intermediate term treasuries issued at time $t$ and maturing at $t + 2\tau$, $T^j_{2\tau,s}$, long term treasuries issued at $t$ and maturing at $t + 3\tau$, $T^j_{3\tau,s}$, and a reference treasury bond maturing at time $t + 4\tau$, $T^j_{4\tau,s}$.[17] The holdings of these securities could be long (asset) or short (liability) depending on the optimal decision of the trading desk.

Basel III has two regulatory constraints dealing with the safety and liquidity of the bank. The newer requirements deal with the liquidity of the bank in both the short term and longer term. The short term is regulated through a liquidity coverage ratio (LCR), which measures the high quality liquid assets to meet one month of unanticipated funding outflow. To represent this requirement, we adopt the liquidity management model of Frost (1970); Freixas and Rochet (2008); and Dutkowsky and VanHoose (2015). Suppose the trading desk manages four marketable securities such that:

$$\sum_{i=1}^{4} T_{i\tau,s} P_{i\tau,s} = \xi K^j_M(s),\tag{18}$$

for $s \in [t, t + \tau]$. Here, $\xi$ is the leverage ratio, so that $1 - \xi$ represents the amount of funds $K^j_M(s)$ invested in the risk free asset. The treasury securities are all zero coupon bonds which trade continuously through time.[18]

The bank faces unanticipated deposit withdrawals for a portion of its deposits. Suppose there is a uniform distribution of deposit flows between two discrete time periods with support $[-\bar{D}, \bar{D}]$. The bank can use its marketable securities as collateral for short-term financing of these deposit withdrawals.[19] If the bank needs to borrow in excess of the net worth of its marketable securities $\xi K^j_M(t)$, then the bank pays a penalty rate $r^p$. The present value of the expected cost of borrowing these funds is

$$C(K^j_M(t)) = r^p \mathcal{M}(\tau, X) \int_{\xi K^j_M(t)}^{\bar{D}} \frac{x - \xi K^j_M(t)}{2\bar{D}} dx = \frac{r^p}{4\bar{D}} \mathcal{M}(\tau, X) \left[ \bar{D} - \xi K^j_M(t) \right]^2.\tag{19}$$

Consequently, the cost of meeting the deposit withdrawals is smaller when the bank holds more marketable securities. Thus, the regulator can modify the liquidity of the bank by restricting the leverage ratio or raising the penalty rate when the bank has to borrow from the central bank.

The longer term liquidity regulation is the net stable funding ratio (NSFR), which is the ratio of available stable funding (ASF) relative to the required stable funding (RSF). In the Section S6.1, we map King (2010) formula for the NSFR into the current banking model from which we get the following constraint:

$$K^j_t \geq \alpha_\tau L^j_{\tau,t} + \alpha_{2\tau} \left( L^j_{2\tau,t} + L^j_{2\tau,t-\tau} \right) + \alpha_T \xi K^j_M(s) - \alpha_K R^j_t.\tag{20}$$

The weights placed on the various categories of funding and assets are given in Table 3. The weight placed on short term loans is less than the one on the longer term loans so that longer term assets lead to a larger increase in RSF. In addition, the weight on government securities is lowest, since these assets are considered more liquid than short term loans, leading to a smaller weight in RSF. Finally, reserves reduce the need for capital since excess reserves can be used to fund liquidity problems.

**Table 3.** Parameters for Regulatory Constraints (20) and (21).

| $\alpha_\tau$ | $\alpha_{2\tau}$ | $\alpha_K$ | $\alpha_T$ | $\kappa_T$ | $\kappa_L$ | $c_b$ |
|---|---|---|---|---|---|---|
| 0.055 | 0.08 | 0.459 | 0.027 | 0.0 | 0.08 | 0.02 |

The risk weighted capital constraint is now:

$$K_s^j \geq \kappa_T \xi K_M^j(s) + \kappa_L \left( L_{\tau,s}^j + L_{2\tau,s}^j + L_{2\tau,s-\tau}^j \right) + c_b \left( \frac{P_{\tau,s}}{\bar{P}_{\tau,s}} - 1 \right)^+, \tag{21}$$

with $\kappa_T < \kappa_L$. Here, $\bar{P}_{\tau,s} = \exp[a_\tau + b_\tau \cdot \bar{X}]$ where $\bar{X}$ is the stationary mean of the state vector. $\kappa_T$ and $\kappa_L$ are the risk weighted capital requirements ratios for treasury securities and loans, respectively. A new item in Basel III is the counter cyclical buffer for all banks $c_b \left( \frac{P_{\tau,s}}{\bar{P}_{\tau,s}} - 1 \right)^+$, where $c_b$ is a positive constant. During good economic times, $P_{\tau,s} > \bar{P}_{\tau,s}$ and the counter cyclical buffer is positive. This corresponds to the level of interest rates below its mean, $X_1(s) < \bar{X}_1$. This counter cyclical buffer does not apply during periods of higher interest rates and lower bond prices.

As in Roelands (2014), there are critical levels of short term loans at the decision time $s = t$ such that the liquidity (20) and capital (21) constraints just bind.

$$L_{l,t}^j = \frac{1}{\alpha_\tau} \left[ K_t^j + \alpha_K R_t^j - \alpha_{2\tau} \left( L_{2\tau,t}^j + L_{2\tau,t-\tau}^j \right) - \alpha_T \xi K_M^j(t) \right], \tag{22}$$

and

$$L_{\kappa,t}^j = \frac{1}{\kappa_L} \left[ K_t^j - \kappa_T \xi K_M^j(t) - \kappa_L \left( L_{2\tau,t}^j + L_{2\tau,t-\tau}^j \right) - c_b \left( \frac{P_{\tau,t}}{\bar{P}_{\tau,t}} - 1 \right)^+ \right]. \tag{23}$$

The treasury securities are all zero coupon bonds which trade continuously through time. The bank's loan and capital decisions are made at discrete intervals and remain fixed within the interval so that time is associated with the beginning of each discrete period, $t$, rather than $s \in [t, t+\tau]$. This means that the Basel III constraints (20) and (21) are not updated within the interval, but imposed by the regulator at the start of every discrete period. If these regulatory constraints applied every instant, then the complexity of the portfolio problem would increase substantially.

*3.2. Counter Cyclical Buffer*

The bank regulator imposes a regulatory cost on the bank based on the state of the financial market. In particular, the regulator wants the bank to hold more capital when the price of financial assets are higher than normal or interest rates are below normal. The purpose is to slow down the expansion of credit, which could be used to fund additional purchases of these assets thus pushing up their prices even further. The counter cyclical buffer (CCB) constraint becomes more binding for the bank in good times, as it forces the bank to raise rates and limit the credit supply. The CCB, in essence, provides insurance to the regulator when market prices heat up. We will show that the CCB can be characterized as a put option that the bank is forced to provide to the regulator. This put option is in-the-money for the regulator when the level of the yield curve is below its mean. From the perspective of the bank, this is a regulatory cost which is conditional on a low level of interest rates.

The critical level of the factors such that the counter cyclical buffer is zero is given by:

$$\rho_b \equiv e^{-A^{\mathcal{P}}(\tau-t)}(\bar{X} - X). \tag{24}$$

The put option characterizing the counter cyclical buffer has a strike price $\bar{X}$ and the payoff is positive when $X < \bar{X}$. The expiration date of the option is the next time period.

The expected cost of the counter cyclical buffer is[20]

$$CCB(X) = c_b \mathcal{M}(2\tau, X) \left( \mathcal{P}(\tau, X) \exp\left\{ \frac{1}{2} b'_\tau \sigma_M(\tau)^{-1} b_\tau \right\} \left[ 1 - \Phi\left( \Sigma_M^{-1}(\rho_b - \sigma_M b_\tau) \right) \right] \right.$$

$$\left. - \left[ 1 - \Phi\left( \Sigma_M^{-1} \rho_b \right) \right] \right), \text{ such that } \frac{\partial CCB(X)}{\partial X} < 0 \text{ for } X > \mu_{\mathcal{M}}. \tag{25}$$

The cutoff (24) leads to the cumulative probability distribution of $Z \in \mathbf{R}^N$ given by

$$\Phi(\rho_b) = \frac{1}{\sqrt{(2\pi)^N}} \int_{\rho_b}^{\infty} e^{-\frac{1}{2} Z'Z} dZ, \text{ such that } \frac{\partial \Phi(\rho_b)}{\partial \rho_b} < 0. \tag{26}$$

$CCB(X)$ is the product of the expected stochastic discount factor $\mathcal{M}(2\tau, X)$ and the expected payoff of this buffer, seen as an option. Here, $\mathcal{P}(\tau, X) = \exp\left\{ b_\tau \left[ e^{-A^{\mathcal{P}}(\tau-t)}(X - \bar{X}) \right] \right\} > 1$ for $X < \bar{X}$, so that the expected payoff is always positive in this case. In addition, the expected payoff is still positive for some $X > \bar{X}$, since there is an adjustment term for risk, $\exp\left\{ \frac{1}{2} b'_\tau \sigma_M(\tau)^{-1} b_\tau \right\}$, which accounts for the uncertainty in the stochastic discount factor.

The expected payoff is multiplied in (25) by the expected stochastic discount factor, which has a normal form for the seller of a put option. As a result, the option value of the counter cyclical buffer for the bank is the mirror image of a normal form in the RHG of Figure 3. Recall that the stochastic discount factor is time varying as it depends on the level of the yield curve. The RHG in Figure 3 allows the level of the yield curve to vary over the interval $X_1 \in [-3\Sigma_{X_1}, 3\Sigma_{X_1}]$. The highest value of the counter cyclical buffer is only 0.018%, since $c_b = 0.02$ and the probability that the buffer applies is small. If the regulator requires additional capital of 1.8% for the counter cyclical buffer, the value of $c_b$ would have to be 100 times bigger.

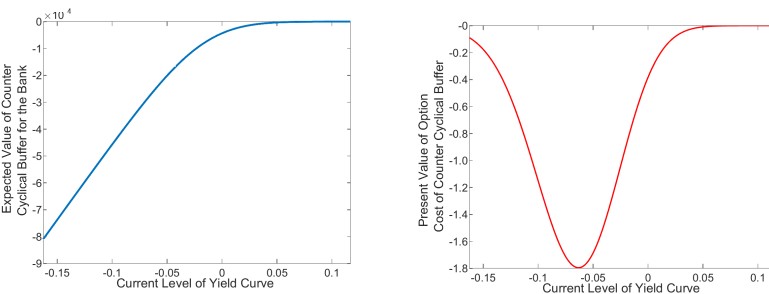

**Figure 3.** The Expected Payoff of Counter Cyclical Buffer (25).

## 4. The Role of the Trading Desk

There has been much discussion in the popular media and in policy circles about the role of rogue traders and their excessive risk taking and leveraging behavior as factors in abetting, if not outright precipitating, the recent financial crisis. Interestingly, there is very little analysis of the trading desk's behavior within the context of a bank or a BHC. This section investigates the role of the trading desk manager's risk attitude as well as leveraging behavior. The analysis sheds light on the behavior of the trading desk manager and its impact on the bank overall risk profile and profitability. By doing so, we hope to help better inform the discussion on bank governance as well as regulatory policy aiming to target trading activities within banks.

Consider, first, the problem of the trading desk manager who is in charge of a trading portfolio for BHC *j* facing interest rate risk, where treasury securities are traded continuously. In contrast, the loan, deposit, and capital decisions are made at discrete times; and these will be explored in the following sections. The capital of bank *j* is composed of two parts, one related to the marketable securities, $K_M^j$, and one related to the loan portfolio, $K_L^j$.

$$K^j = K_L^j + K_M^j. \tag{27}$$

The marketable securities are invested by the trading desk following the regulatory constraint (18). Consequently, the return on these securities of a specific maturity follows the stochastic process (12) which the bank takes as given. The trading desk is allowed to re-balance these funds throughout the time period $s \in [t, t+\tau]$. We use four securities namely 3 month, 2, 5, and 10 year bonds, since the constraint (18) reduces the number of independent choices to the number of factors, 3.

To integrate the trading desk's problem into the overall problem for the bank we define the change in the trading desk's capital by

$$dK_M^j(s) = \pi_M^j(s)ds + \sigma_\pi d\epsilon_s. \tag{28}$$

The profits of the trading desk at each instant are

$$\pi_M^j(s) \equiv (1 - \xi)r(s)K_M^j(s) + \sum_{i=1}^{4} \mu_{i\tau}(s)T_{i\tau,s}P_{i\tau,s},$$

subject to (18). The instantaneous expected excess rates of return on marketable securities, from (12), are

$$\mu_{i\tau}(s) - r(s) \equiv b_{i\tau}'\left[(\gamma^{\mathcal{P}} - \gamma^{\mathcal{Q}}) - (A^{\mathcal{P}} - A^{\mathcal{Q}})X(s)\right], i = 1, 2, 3, 4.$$

There is also a volatility component of profits earned by bank *j* at any time $s \in [t, t+\tau]$ given by

$$\sigma_\pi \equiv \left[T_{\tau,s}^j P_{\tau,s} b_\tau + T_{2\tau,s}^j P_{2\tau,s} b_{2\tau} + T_{3\tau,s}^j P_{3\tau,s} b_{3\tau} + T_{4\tau,s}^j P_{4\tau,s} b_{4\tau}\right]\Sigma_X. \tag{29}$$

In order to highlight the role of attitude toward risk in affecting the type of investments made and implications for the rest of the bank, the trading desk manager *j* is assumed to be risk averse with a constant relative risk aversion utility (CRRA) with parameter $\gamma^j$. We can now specify the portfolio problem of the trading desk manager for bank *j*, which is to maximize the expected utility from terminal capital at a fixed time $\tau$ given its current market capital, $K_M^j(t) = K_M^j$ and interest rate factors, $X(t) = X$. The trading desk of bank *j* has an investment horizon $\tau$. The bank's conditional expected value related to the actions of the trading desk manager is

$$J(K_M^j, X, \tau, t) = e^{-\beta\tau}E\left[\frac{\left(K_M^j(\tau)\right)^{1-\gamma^j}}{1-\gamma^j}\middle| K_M^j(t) = K_M^j, X(t) = X\right], \tag{30}$$

where $\beta$ is the discount rate for the bank.

The bank capital, $K_M^j$, associated with the marketable securities in the bank's portfolio follows

$$\frac{dK_M^j(s)}{K_M^j(s)} = \left[(1-\xi)r(s) + \omega(s)'\mu(s) + \omega_4(s)\mu_{4\tau}(s)\right]ds + \omega(s)'b\Sigma_X d\epsilon_s + \omega_4(s)b_{4\tau}\Sigma_X d\epsilon_s$$

for $s \in [t, t+\tau]$, and $\omega(s)'\iota + \omega_{4\tau}(s) = \xi$, where the weights are now defined as:

$$\omega(s)' \equiv \left[T_{\tau,s}^j P_{\tau,s}, T_{2\tau,s}^j P_{2\tau,s}, T_{3\tau,s}^j P_{3\tau,s}\right]/K_M^j(s), \omega_{4\tau}(s) = T_{4\tau,s}^j P_{4\tau,s}/K_M^j(s) \tag{31}$$

$$\mu(s) \equiv [\mu_\tau(s), \mu_{2\tau}(s), \mu_{3\tau}(s)],$$

with $\mu_{i\tau}(s) \equiv r(s) + b_{i\tau}'\left[(\gamma^{\mathcal{P}} - \gamma^{\mathcal{Q}}) - (A^{\mathcal{P}} - A^{\mathcal{Q}})X(s)\right], i = 1, 2, 3, 4.$

Here $b' = \begin{pmatrix} b_\tau & b_{2\tau} & b_{3\tau} \end{pmatrix}$.

The trading desk's problem has been solved by Sangvinatsos and Wachter (2005) and Liu (2007). They find that the value function for the trading desk manager is[21].

$$J(K_M^j, X, \tau, t) = \frac{\left(K_M^j(t)\right)^{1-\gamma^j}}{1-\gamma^j} J(\tau, X), \tag{32}$$

$$\text{where } J(\tau, X) = J(\tau) \exp\left\{-\frac{1}{2}(X - \mu_J(\tau))'(\sigma_J(\tau))^{-1}(X - \mu_J(\tau))\right\}^{\gamma^j}.$$

Given the solution, the portfolio rule for the trading desk is given by:

$$\omega(t) = \omega_1\left\{(b - \iota b_{4\tau})\left[(\gamma^{\mathcal{P}} - \gamma^{\mathcal{Q}}) - (A^{\mathcal{P}} - A^{\mathcal{Q}})X(t)\right]\right\} + \omega_2\xi + \omega_3\gamma^j(\sigma_J(\tau))^{-1}[X - \mu_J(\tau)]$$

$$\omega_1 \equiv \left[\gamma^j\left(b\Sigma_X\Sigma_X'b' + \iota'b_{4\tau}\Sigma_X\Sigma_X'b_{4\tau}' - 2b\Sigma_X\Sigma_X'b_{4\tau}'\iota'\right)\right]^{-1} \text{ with } \iota' = \begin{pmatrix} 1 & 1 & 1 \end{pmatrix}, \tag{33}$$

$$\omega_2 \equiv 2\omega_1\left(b\Sigma_X\Sigma_X'b_{4\tau}' - \iota b_{4\tau}\Sigma_X\Sigma_X'b_{4\tau}'\right) \text{ and } \omega_3 \equiv \omega_1(b - \iota b_{4\tau})\Sigma_X\Sigma_X'.$$

$$\omega_4(t) = \xi - \iota'\omega(t).$$

The first term in the portfolio rule is the traditional Sharpe ratio adjusted for risk $\gamma^j$, since the expected excess return on the treasury securities is $(b - \iota b_{4\tau})[(\gamma^{\mathcal{P}} - \gamma^{\mathcal{Q}}) - (A^{\mathcal{P}} - A^{\mathcal{Q}})X(t)]$ and the variance-covariance matrix $b\Sigma_X\Sigma_X'b' + \iota'b_{4\tau}\Sigma_X\Sigma_X'b_{4\tau}' - 2b\Sigma_X\Sigma_X'b_{4\tau}'\iota'$ from (12) determines $\omega_1$. However, the excess return is measured relative to the 4th asset. Consequently, the price of risk $(\gamma - \gamma^{\mathcal{Q}}) - (A - A^{\mathcal{Q}})X(t)$ is multiplied by the elasticity of the bond with maturity $1, 2$ or $3$ minus the elasticity for the $4^{th}$ bond, $b - \iota b_{4\tau}$. In addition, the variance-covariance of the first three bonds, $b\Sigma_X\Sigma_X'b'$, is adjusted for the variance of the fourth asset, $b_{4\tau}\Sigma_X\Sigma_X'b_{4\tau}'$ and the covariance of the three assets with the fourth asset, $b\Sigma_X\Sigma_X'b_{4\tau}'$. The second term is an adjustment to ensure that the portfolio weights add up to $\xi$.

The last term in the portfolio rule (33) is the hedging demand for treasury securities from Merton (1971). This term consists of the regression coefficients (beta) for the excess returns on treasury securities against the interest rate factors, $\omega_3$, and the sensitivity of the expected lifetime utility with respect to the factors $\gamma^j(\sigma_J(\tau))^{-1}[X - \mu_J(\tau)]$. This latter term can be interpreted as the risk adjusted duration of bank $j$. Table 4 provides the key parameters for the lifetime utility of the trading desk with an investment horizon of 1 year, coefficient of relative risk aversion $\gamma^j = 10$, discount rate $\beta = 0.05$, and leverage ratio $\xi = 1$. The graphs use only the level of the yield curve factor so that the graphs are two dimensional. As a result, the investor has only two independent bonds to invest in. The lifetime utility is for an investor that invests in 3 months, and 5 year bonds using the estimated parameters for the term structure from section II.A. LHG of Figure 4 gives this lifetime utility for the level of the yield curve $X_1 \in [-3\sigma_J(\tau) + \mu_J(\tau), 3\sigma_J(\tau) + \mu_J(\tau)]$. Consequently, it is possible for the level of the lifetime utility curve to be above or below the mean of the lifetime utility, so that the hedging demand can be positive or negative,

respectively. This result is shown in the RHG in Figure 4. The hedging demand (red dotted line) is zero at the mean of the expected lifetime utility.

**Table 4.** Solution to the Lifetime Utility of the trading desk manager.

| $\gamma^j$ | $\beta$ | $J(\tau)$ | $\mu_J(\tau)$ | $\sigma_J(\tau)$ |
|---|---|---|---|---|
| 10 | 0.05 | 0.9757 | $-0.0593$ | 0.1065 |

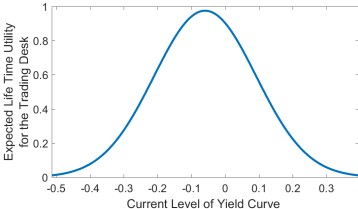 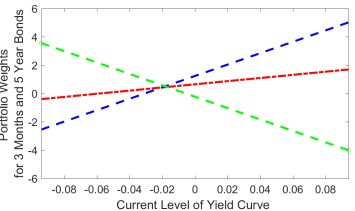

**Figure 4.** The Expected Lifetime Utility of the trading desk manager (32) and PortfolioWeights (33).

For $X > \bar{X}$, mean reversion (4) implies that the trading desk manager expects the level of the yield curve to fall, and hence longer duration bonds would lead to a larger capital gain. As a result, the trading desk is long five year bonds (blue dashed line) and short three month bonds (green short-dashed line) for a high expected level of the yield curve, $X_1 > \bar{X}_1 = -0.0177$. If the random change in the future yield factors is positive, $Y_\tau > 0$, then the trading desk would suffer a large capital loss. The trading desk's position is reversed for lower levels of the yield curve, $X_1 < \bar{X}_1$, since mean reversion implies that the trading desk expects the level of the yield curve to move back to its stationary value by (4). If the random change in future yield factors is negative, then the trading desk would suffer a capital loss.

We can now calculate the impact of the trading desk manager's investment behavior on his conditional expected gross growth rate of capital, given the stochastic process for the term structure factors (3).[22]

$$E_t\left(\frac{K_M^j(t+\tau)}{K_M^j(t)}\right) \equiv \mathcal{K}(\tau, X) = \mathcal{K}(\tau)\exp\left\{-\frac{1}{2}\left(X - \mu_{\mathcal{K}}(\tau)\right)'\left(\sigma_{\mathcal{K}}(\tau)\right)^{-1}\left(X - \mu_{\mathcal{K}}(\tau)\right)\right\}, \tag{34}$$

where $\mathcal{K}(\tau)$, $\mu_{\mathcal{K}}$ and $\sigma_{\mathcal{K}}$ are derived in the Section S1. These parameters are given in Table 5 for the trading desk characterized in Table 4. Note that Equation (34) is deterministic and conditional on information at time $t$, i.e., $X_t = X$.

**Table 5.** Mean and Standard Deviation for Expected and Unexpected Gross Growth Rate of Trading Desk Manager's Capital.

| $\gamma^j$ | $\mathcal{K}(\tau)$ | $\mu_{\mathcal{K}}(\tau)$ | $\sigma_{\mathcal{K}}(\tau)$ | $\sigma_K(\tau)$ | $\bar{X}_1 - \Sigma_{X_1}$ | $\bar{X}_1$ | $\bar{X}_1 + \Sigma_{X_1}$ |
|---|---|---|---|---|---|---|---|
| 10 | 1.2138 | $-0.0639$ | 0.1065 | 0.0104 | 0.0934 | 0.0054 | 0.2640 |
| 5 | 1.0584 | $-0.0567$ | 0.1167 | 0.0105 | 0.2030 | 0.0085 | 0.4672 |

*The Role of the Attitude toward Risk*

In Figure 5 (top LHG), the conditional expected gross growth rate of the trading desk's capital (34) is plotted against the first factor for the term structure for $\gamma^j = 10$. Figure 5 (top RHG) plots the same graph for $\gamma^j = 5$. At the stationary value for the level of the yield curve $\bar{X}_1 = -0.0177$, we have that $\mathcal{K}(\tau, X) = 1.1048$ for a time horizon of one year and $\gamma^j = 10$, so that the expected growth rate is 10.48% under normal circumstances. However, there can be a substantial capital loss of 18.52% when the level of the yield curve reaches its maximum observed value of 0.0256, as calculated using (34). If the trading desk has 20% of the bank's capital, then the total capital of the bank could fall by 3.70%.

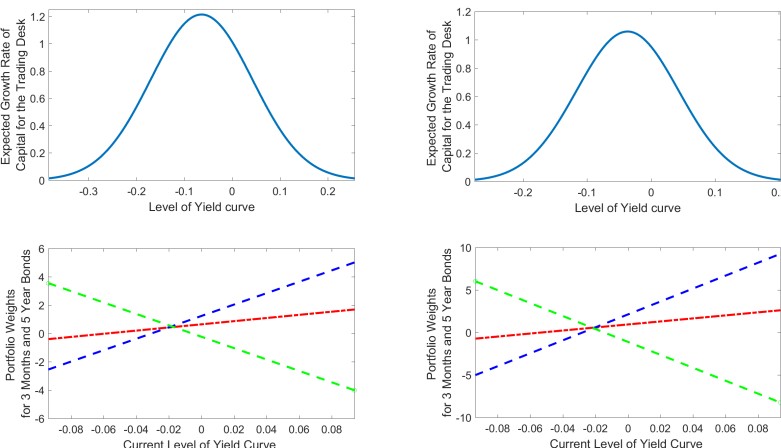

**Figure 5.** Gross Rate of Return for the Trading Desk Manager's Capital, $\gamma^j = 10$ (LHG) and $\gamma^j = 5$ (RHG).

The risk averse trading desk hedges against the mean reversion of the level of the yield curve by going long (short) in 5 year government securities for a high (low) level of the yield curve. In the bottom LHG of Figure 5 we reproduce the portfolio decision for $\gamma^j = 10$, while the bottom RHG corresponds to $\gamma^j = 5$. As depicted in these two graphs, the trading desk manager with a lower aversion to risk increases the magnitude of the bet that the level of the yield curve will revert to its long term mean. This implies that the less risk averse trading desk manager will choose a portfolio with higher duration, relative to the manager that is more risk averse. Interestingly, and perhaps, initially, more surprising, the expected gross growth rate of capital for the less risk averse trading desk manager is lower than that of the more risk averse manager. In other words, the more risk averse trading desk manager is providing more value to the bank than the more risk aggressive manager. How is that possible? The answer is that, first, the higher duration portfolio of the less risk averse manager is more susceptible to interest rate volatility, and, second, the no arbitrage condition rules out profiting from such volatility. As a result, the expected gross growth rate of capital for the more risk averse manager is higher.

Moreover, the convexity correction, due to interest rate volatility, is higher for the higher duration portfolio, which lowers its expected value. As a result, and given the inverse relationship between price and return, the expected holding period return for that portfolio will be higher. This can be observed in the last three columns of Table 5, which provide the holding period return at the stationary value of the level of the yield curve and this level plus or minus one standard deviation. At all levels of the yield curve, we see that the current expected holding period return on the portfolio is larger for a lower aversion to risk.

Next, we explore the role of the random component for the gross growth rate of capital for the trading desk. Using the forward Kolmogorov equation, we calculate the transitional probability $p_K(t, X, \tau, Y)$ from the state $X$ at time $t$ to the state $Y$ at time $t + \tau$:[23]

$$p_K(t, X, \tau, Y) = \frac{\exp\left\{ -\frac{1}{2} Y' \sigma_K(\tau)^{-1} Y \right\}}{\sqrt{(2\pi)^N det(\sigma_K(\tau))}}. \tag{35}$$

This probability distribution represents the capital loss when $Y > 0$, since the future level of the yield curve is above what the trading desk expected.

We can therefore write

$$\frac{K_M^j(t + \tau)}{K_M^j(t)} = \mathcal{K}(\tau, X) p_K(t, X, \tau, Y). \tag{36}$$

Given the estimates of the term structure model and using a one year horizon, we find that the trading desk's capital has $\sigma_K(\tau) = 0.0104$ given the trading desk follows the optimal portfolio rules (33) with CRRA $\gamma^j = 10$, $\beta = 0.05$, and $\xi = 1$.

The partition of the growth rate of capital for the trading desk into deterministic and random components is useful for various evaluations of the performance of the bank. First, we can now calculate the expected cash flow from the trading desk to the overall bank using (33) and (35).[24]

$$E_t\left[\exp\left\{\int_t^{t+\tau}\left[(1-\xi)r(s) + \omega^*(s)'\mu(s) + \omega_4^*(s)\mu_{4\tau}(s)\right]ds + \sigma_\pi\int_t^{t+\tau}d\epsilon_s\right\}\right] = \mathcal{K}(\tau, X). \tag{37}$$

The market valuation of the cash flows generated by the trading desk uses the stochastic discount factor given by (10). The present value of the marginal value of the trading desk's capital is also a function of the gross growth rate of the trading desk's capital, and is given by:

$$\frac{M_{\tau,t}}{M_{t,t}}\frac{K_M^j(t+\tau)}{K_M^j(t)} = \mathcal{M}(\tau, X)\mathcal{K}(\tau, X)\frac{\exp\left\{-\frac{1}{2}Y'\left(\sigma_M(\tau)^{-1} + \sigma_K(\tau)^{-1}\right)Y\right\}}{\sqrt{(2\pi)^N \det\left[\left(\sigma_M(\tau)^{-1} + \sigma_K(\tau)^{-1}\right)^{-1}\right]}}. \tag{38}$$

The conditional distribution $\mathcal{M}(\tau, X)\mathcal{K}(\tau, X)$ of the current yield factors, $X$, has a Gaussian form with parameters $\mu_{\mathcal{MK}}(\tau) = -0.0294$ and $\sigma_{\mathcal{MK}}(\tau) = 0.0427$. The parameters of the distribution of the random changes in the yield curve factors $Y$ are given in Table 6. This distribution still has mean zero but the variance is larger relative to the stochastic discount factor (10) and the gross growth rate of the trading desk's capital (36).

**Table 6.** Mean and Standard Deviation for the Discounted Expected and Unexpected Gross Growth Rate of Trading Desk Manager's Capital.

| $\mathcal{M}(\tau)\mathcal{K}(\tau)$ | $\mu_{\mathcal{MK}}(\tau)$ | $\sigma_{\mathcal{MK}}(\tau)$ | $\left(\sigma_M(\tau)^{-1} + \sigma_K(\tau)^{-1}\right)^{-1}$ |
|:---:|:---:|:---:|:---:|
| 1.2215 | $-0.0294$ | 0.0427 | 0.0106 |

Thus, the present value of the expected cash flow for the trading desk's capital (38) is determined by his starting capital allocation $K_M^j(t)$, the deterministic component of the growth rate of the trading desk's capital (34), and the expected stochastic discount factor over the trading desk's time horizon.[25]

## 5. The Role of the COO

In this section, we will focus on the lending business of the BHC, and then discuss how it is affected by the trading desk decisions. We denote by $\pi_L^j(s)$ the profits of the lending desk (to be defined in the next section). As discussed earlier, the COO has the option to raise additional capital $I_{\tau,t}^j = \int_t^{t+\tau} q^j ds = q^j\tau$ at the beginning of the period. This is done continuously over the period at the constant rate $q^j$. The COO may also choose to pay dividends at the constant rate $r_{\tau,t}^{jK}$ over the period $[t, t+\tau]$. Consequently, the evolution of the bank's capital is

$$dK^j(s) = \left[\pi_L^j(s) - r_{\tau,t}^{jK} + q^j\right]ds. \tag{39}$$

The change in capital for the bank over the horizon $t$ to $t + \tau$ is

$$K^j(t+\tau) - K^j(t) = \left[\pi_L^j(t) - r_{\tau,t}^{jK} + q^j\right]\tau. \tag{40}$$

Given the current capital of the bank, the COO has to allocate it between the trading desk and the loan desk so that:

$$K^j(t) = K_M^j(t) + K_L^j(t).$$

The COO takes the trading desk's balance sheet constraint (18) as given, so that the balance sheet of the bank (17) reduces to:

$$R_t^j + L_{\tau,t}^j + L_{2\tau,t}^j + L_{2\tau,t-\tau}^j = D_t^j + K_L^j(t) + (1 - \xi)K_M^j(t). \tag{41}$$

The value of the bank consists of the sum of the values of the lending and trading businesses. As discussed earlier, the market stochastic discount factor (10) is used to price all cash flows. The bank's objective is:[26]

$$V\left(t, K_M^j(t), K_L^j(t), L_{2\tau,t-\tau}^j, r_{2\tau,t-\tau}^j, X(t)\right) = \max_{q^j, r_{\tau,t}^{jK}, K_M^j(t)} \mathcal{M}(\tau, X) \left\{ K_M^j(t)\mathcal{K}(\tau, X) \right.$$

$$- \frac{r^p}{4\bar{D}}\left[\bar{D} - \xi K_M^j(t)\right]^2 + \max_{r_{\tau,t}^j, r_{2\tau,t}^j, K_L^j(t+\tau)} E_t \left\{ \left[\pi_L^j(t) - (1-\chi)r_{\tau,t}^{jK} - r_{\tau,t}^D D_t^j + (1-\eta)q^j\right]\tau \right. \tag{42}$$

$$\left. \left. + \frac{\mathcal{M}(2\tau, X)}{\mathcal{M}(\tau, X)}E_t\left[p_M(2\tau, Y)V\left(t+\tau, K_M^j(t+\tau), K_L^j(t+\tau), L_{2\tau,t}^j, r_{2\tau,t}^j, X(t+\tau)\right)\right] \right\} \right\}.$$

The first term in (42) is the expected value of the discounted cash flows generated by the trading desk from $t$ to $t + \tau$. Each cash flow over the period $t$ to $t + \tau$ is the gross growth rate of capital for the trading desk which is discounted using the stochastic discount factor (10) for $s \in [t, t + \tau]$. Consequently, the present value of the expected cash flows of the trading desk is given by (38) for any time period $s \in [t, t + \tau]$.

The Bank COO knows $\Omega_{t,\tau} = \left\{\varepsilon_{\tau,t}^j, \varepsilon_{2\tau,t}^j, \mathcal{M}(\tau, X)\mathcal{K}(\tau, X), p_K(t, X, \tau, Y)\right\}$ at time $t$. Note that the marginal cost of raising net new capital $\eta$ reflects the fact that a seasoned offering of new shares is costly. On the other hand, the marginal benefit to the bank, $\chi > 1$, accounts for the benefit to shareholders of regular dividend payments. The bank COO chooses $\left\{r_{\tau,t}^j, r_{2\tau,t}^j, K_L^j(t+\tau), r_{\tau,t}^{jK}, q^j\right\}$ subject to the regulatory capital constraint (21) with Lagrange multiplier $\lambda_1(t)$, the net stable funding constraint (20) with Lagrange multiplier $\lambda_2(t)$, and its balance sheet constraint (41).

The cost of seasoned equity offerings and the benefit of initiating dividend payments have been examined in financial economics.[27] Gao and Ritter (2010) examine seasonal equity offerings which arise from the effort of the investment banker to improve the elasticity of demand for the corporation's shares. This effort, however, would be limited in a financial crisis. Following Corwin (2003) and Mola and Loughran (2004), among others, the cost of the offering depends on, both, an explicit fee from the investment bank and an underpricing of the offering. In addition, the cost of the offering escalates as the offering increases. Starting with Michaely et al. (1995), the benefit from the payment of dividends has been measured by either the initiation of dividends or their omission. They find a gain of about 3% from the initiation of dividends and −7% from the suspension of dividend payments. Bessembinder and Zhang (2013) examine the impact of these corporate events on the long term return on corporate stocks. In both cases they confirm previous results and find that the impact is dependent on the corporate environment including change in momentum, illiquidity, and idiosyncratic volatility. Consequently, the values of these parameters are dictated by the specific characteristics of the corporation including whether or not the firm is in financial crisis. For the purpose of our exercise, we take the cost or benefit to be 7%.

The COO issues new equity based on:

$$\mathcal{M}(\tau, X)(1 - \eta)\tau + \mathcal{M}(2\tau, X)E_t\left[p_M(2\tau, Y)\frac{\partial V}{\partial K_L^j(t + \tau)}\right] \le 0. \tag{43}$$

If this inequality is true, then the bank does not issue equity.

The COO's decision to pay dividends is based on:

$$\mathcal{M}(\tau, X)(\chi - 1)\tau - \mathcal{M}(2\tau, X)E_t\left[p_M(2\tau, Y)\frac{\partial V}{\partial K_L^j(t + \tau)}\right] \le 0. \tag{44}$$

If this inequality is true, then the bank does not pay dividends.[28]

The COO's first order condition to allocate the capital of the bank is conditional on whether $K_M^j(t)$ is at the lowest value of zero or the highest possible value $K^j(t)$.

$$\frac{\partial V}{\partial K_M^j(t)} - \frac{\partial V}{\partial K_L^j(t)} \le 0 \text{ for } K_M^j(t) = 0 \tag{45}$$

or

$$\frac{\partial V}{\partial K_M^j(t)} - \frac{\partial V}{\partial K_L^j(t)} \ge 0 \text{ for } K_M^j(t) = K^j(t). \tag{46}$$

If the value of the trading desk's capital is between these two extremes, then $K_M^j(t)$ is chosen such that:

$$\frac{\partial V}{\partial K_M^j(t)} = \frac{\partial V}{\partial K_L^j(t)}. \tag{47}$$

### 5.1. Determining the Optimal Loan Margin

In this section, we will analyze the loan decision of the bank, given the trading desk's position and in the presence of capital and liquidity constraints. The bank COO has to choose the interest rate margin for loans, $L$, with two maturities, $\tau \in \{\tau, 2\tau\}$. For each maturity, the value of total loan demand for bank $j$ is given by:

$$L_{\tau,t}^{d,j} = \gamma_{0,\tau}^j - \gamma_{1,\tau}^j r_{\tau,t}^j + \sigma(r_{\tau,t}^j)\varepsilon_{\tau,t}^j \text{ for } \tau \text{ either } \tau, \text{ or } 2\tau, \tag{48}$$

where $r_{\tau,t}^j$ is the interest rate charged on loans of maturity either $\tau$, or $2\tau$. Following the banking literature, borrowers undertake riskier projects when interest rates increase.[29] Specifically, $\frac{d\sigma(r_{\tau,t}^j)}{dr_{\tau,t}^j} > 0$ with $\varepsilon_{\tau,t}^j = g$ or $b$, with $p$ being the probability of a good outcome, $g$.[30] In particular, the standard deviation of this shock is

$$\sigma(r_{\tau,t}^j) = \sigma_0 + \sigma_1 r_{\tau,t}^j > 0 \Rightarrow \sigma'(r_{\tau,t}^j) = \sigma_1 > 0. \tag{49}$$

If the bank commits one unit to loans or the safe asset, the shocks satisfy

$$b < \frac{L_{\tau,t}^{d,j} - (\gamma_{0,\tau}^j - \gamma_{1,\tau}^j r_{\tau,t}^j)}{\sigma_1 r_{\tau,t}^j + \sigma_0} < g.$$

Consequently, bank $j$ has the largest marginal revenue in the good state, which is larger than the return from the safe asset. In addition, the marginal revenue from the loan in the bad state is worse than the benefit from the safe asset.

The interest rate charged by bank $j$ on its loans consists of two components: the interest rate on U. S. Treasuries of the appropriate maturity, $r_{\tau,t}$, and the interest rate margin, $m_{\tau,t}^j$, which follows from market power of the bank in the loan market for each bank. We have:

$$r_{\tau,t}^j = r_{\tau,t} + m_{\tau,t}^j \text{ for } j = 1, \cdots, N. \tag{50}$$

The interest rate margin will be set relative to the marginal cost of loans $c^j$. This marginal cost $c^j = 0.0378$ is given in Table 7. It is set in Section S6.2 of the Supplementary using the Call Reports for the 500 largest commercial banks in the U. S. from Quarter I of 2001 to Quarter IV of 2007.[31] This parameter is set equal to the average of the ratio of non-interest expenses relative to total assets, across all banks and time.

**Table 7.** Parameters for Deposits (51) and Reserves (52).

| $c^j$ | $d_0$ | $d_1$ | $r^p$ | $\bar{D}$ | $r_0$ | $r_1$ |
|-------|-------|-------|-------|-----------|-------|-------|
| 0.0378 | 0.0111 | 0.0282 | 0.1 | 0.0022 | 0.1340 | 0.3936 |

To close the loan desk problem, we specify a model for the interest rate paid on deposits, $r_\tau^D$, and reserve balances, $R^j$. In the simulations we specify that these variables are linear functions of the level of the yield curve, X(t). To have a sense about how the level of the yield curve would, on average across banks, affect the interest rates paid on deposits and reserve balances, we run panel regressions with fixed bank effects using bank specific data from the same Call Reports.[32]

The bank's deposit rate would in general be lower than the corresponding yield to maturity on government securities, such that the bank can still cover the marginal cost of providing deposits.[33] As a result, we assume that

$$r_{\tau,t}^D = d_0 + d_1 X(t). \tag{51}$$

The interest rate factors determine the yields to maturity so that the constant $d_1$ is related to the coefficients in (1), while the constant $d_0$ is related to the marginal cost of deposits.

To set the parameters for the deposit rate in Table 7, we estimate a linear regression of the interest expenses on deposits to deposit ratio on the first latent variable from the term structure estimates. A panel regression with bank fixed effects is estimated using the same Call Report data. The statistically significant parameters are recorded in Table 7.

Dutkowsky and VanHoose (2015) provide a model of reserve holding in the face of interest payment on reserves. They find that the optimal holding of reserves under an interior solution is dependent on the other interest rates in the economy, which are related to the interest rate factors. The central bank would have to set the interest on reserves so that the demand for reserves is equal to the amount the central bank wants to supply. This implies that the interest rate on reserves would also have to be dependent on the interest rate factors. Consequently, we assume reserves are related to the interest rate factors.

$$R_t^j = r_0 + r_1 X(t). \tag{52}$$

Here the constant $r_1$ is dependent on the marginal cost of the various assets and liabilities of the bank and the coefficients in (1). The constant $r_0$ would be related to the amount of reserves the central bank wants within the banking system.

The parameters for the model for reserve holdings are given in Table 7. In this case, a panel regression with bank fixed effects is estimated using cash balances plus deposits due from other depository institutions as the dependent variable and the first latent variable from the term structure estimates is the independent variable. The panel of banks is the same as for the deposit rate regression. The parameters are the statistically significant estimates of the coefficients from this panel regression. In Table 7, we assume a penalty

rate of 10% and the maximum deposit withdraws to be 0.22% of the bank's assets, which determine the cost of the short term liquidity constraint (19) under Basel III.

We now consider the loan problem given the solution to the trading desk's problem. The profit from the loan portfolio is:

$$\pi_L^j(s) = (r_{\tau,t}^j - c^j)L_{\tau,t}^j + (r_{2\tau,t}^j - c^j)L_{2\tau,t}^j + (r_{2\tau,t-\tau}^j - c^j)L_{2\tau,t-\tau}^j - r_{\tau,t}^D D_t^j \text{ for } s \in [t, t+\tau].$$
(53)

If the COO pays dividends, then (44) is an equality, so that the first order condition for choosing the loan margin is:

$$\chi \frac{\partial \pi_L^j}{\partial m_{\tau,t}^j} = [\lambda_1 \kappa_L + \lambda_2 \alpha_\tau]\left(\gamma_{1,\tau}^j - \sigma_1 \varepsilon_{\tau,t}^j\right).$$
(54)

Here,

$$\frac{\partial \pi_L^j}{\partial m_{\tau,t}^j} = \left[2\left(-\gamma_{1,\tau}^j + \sigma_1 \varepsilon_{\tau,t}^j\right)r_{\tau,t}^j - \left(c^j + r_{\tau,t}^D\right)\left(-\gamma_{1,\tau}^j + \sigma_1 \varepsilon_{\tau,t}^j\right) + \gamma_{0,\tau}^j + \sigma_0 \varepsilon_{\tau,t}^j\right]\tau.$$

If the bank issues equity, then $\chi$ is replaced by $\eta$.

If regulatory constraints (20) and (21) are not binding, then the COO equates marginal revenue with marginal cost so that the first order condition (54) becomes:

$$m_{\tau,t}^{j*} = \frac{1}{2}\left(c^j + r_{\tau,t}^D\right) - \frac{\gamma_{0,\tau}^j + \sigma_0 \varepsilon_{\tau,t}^j}{2\left(-\gamma_{1,\tau}^j + \sigma_1 \varepsilon_{\tau,t}^j\right)} - r_{\tau,t}(X).$$
(55)

In this case, $\lambda_1(t) = 0$ and $\lambda_2(t) = 0$. The $'*'$ refers to the unconstrained solution. If this interest rate margin is substituted into the demand for loans, we get the loan level $L^*$ when the capital or liquidity constraints are not binding. By construction, the COO sets the margin such that the loan rate covers the marginal cost of bank liabilities. In addition, there is a surcharge for the monopoly power of the bank based on the elasticity of the loan demand with respect to a change in the loan rate.

We now consider the optimal loan margin of the bank when the capital constraint is binding, so that we can identify the expected marginal value of bank capital (see Chami and Cosimano (2010)). When $L_{\kappa,t}^j < L^*$, the capital constraint (23) is binding and $\lambda_1^*(t) > 0$. The loan rate will be determined by the demand for loans (48). The loan margin is

$$m_{\tau,t}^{j\kappa} = \frac{1}{\left(-\gamma_{1,\tau}^j + \sigma_1 \varepsilon_{\tau,t}^j\right)}\left[-\left(\gamma_{0,\tau}^j + \sigma_0 \varepsilon_{\tau,t}^j\right) + L_{\kappa,t}^j\right] - r_{\tau,t}(X).$$
(56)

The superscript $'\kappa'$ in the loan margin refers to the loan margin under the capital constraint (21).

Given the optimal loan margin, we can use (54) to find the Lagrange multiplier for the liquidity constraint.

$$\lambda_1^*(t) = 2\tau \frac{\chi}{\kappa_L}\left[r_{\tau,t}^{j\kappa} - \frac{1}{2}\left(c^j + r_{\tau,t}^D\right) + \frac{\gamma_{0,\tau}^j + \sigma_0 \varepsilon_{\tau,t}^j}{2\left(-\gamma_{1,\tau}^j + \sigma_1 \varepsilon_{\tau,t}^j\right)}\right]$$
$$= 2\tau \frac{\chi}{\kappa_L}\left[r_{\tau,t}^{j\kappa} - r_{\tau,t}^{j*}\right].$$
(57)

Suppose the current level of bank capital is $K_t^j$, which would correspond to a regulatory level of loans, $L_{\kappa,t}^j$, (21), and loan rate from (56), $r_{\tau,t}^{j\kappa}$. Now, if the bank finds it optimal to choose the unconstrained loan rate $r_{1,t}^{j*} > r_{1,t}^{j\kappa}$, then the optimal level of loans is $L_{\tau,t}^{j*} < L_{\kappa,t}^j$

so that $K_t^{j*} < K_t^j$. In this case, the capital constraint does not bind and the Lagrange multiplier is zero, since the bank's choice of capital is inside the capital constraint. As a result, we have:

$$\lambda_1^*(t) = \begin{cases} 2\tau \frac{\chi}{\kappa_L} \left[ r_{\tau,t}^{j\kappa} - r_{\tau,t}^{j*} \right] & \text{for } r_{\tau,t}^{j\kappa} > r_{\tau,t}^{j*} \\ 0 & \text{for } r_{\tau,t}^{j\kappa} \leq r_{\tau,t}^{j*}. \end{cases} \tag{58}$$

When the liquidity constraint is binding $\frac{\chi}{\kappa_L}$ is replaced by $\frac{\chi}{\alpha_\tau}$ and the loans are binding at rate $r_{\tau,t}^{jl}$, so that[34]

$$\lambda_2^*(t) = \begin{cases} 2\tau \frac{\chi}{\alpha_\tau} \left[ r_{\tau,t}^{jl} - r_{\tau,t}^{j*} \right] & \text{for } r_{\tau,t}^{jl} > r_{\tau,t}^{j*} \\ 0 & \text{for } r_{\tau,t}^{jl} \leq r_{\tau,t}^{j*}. \end{cases} \tag{59}$$

In the rest of the paper, we calibrate the parameters for the bank's loan demand to those in Table 8. Specifically, we use the first order conditions for loans (55) together with the sample average of key financial ratios across the largest 500 commercial banks in U.S. and the time period, Quarter I of 2001 to Quarter IV of 2007. These ratios include the average interest expenses on deposits relative to total deposits, $r^D = 0.0165$ per year, the average ratio of interest and fees on commercial and industrial loans to total commercial and industrial loans, $r^j = 0.0643$, per year and the ratio of the average non-interest expenses to total assets $c^j = 0.0376$ per year across all 500 commercial banks from 2001 to 2007. Using (55) under $\varepsilon_{\tau,t}^j = 0$, we can set the ratio of the constant to slope from the demand for loans (48), without uncertainty, at $\frac{\gamma_{0,\tau}^j}{2\gamma_{1,\tau}^j} = 0.0372$. This result implies an elasticity of demand of 6.4. We then use the demand for loans without uncertainty (48) to set its constant and slope in Table 8, so that the demand for loans is the average value of commercial and industrial loans relative to total assets across all 500 commercial banks and time.

**Table 8.** Parameters for Loan Demand (48) and (49).

| $\gamma_{0,\tau}$ | $\gamma_{1,\tau}$ | $\sigma_0$ | $\sigma_1$ | $z_0$ | $z_1$ |
|---|---|---|---|---|---|
| 0.8972 | 12.0621 | 0.0331 | 0.2067 | −0.6150 | 0.00035 |

To set the parameters for the uncertainty in loan demand (48), we also use information from the 500 largest commercial banks: $\sigma_0 = 0.0331$, is the standard error from a panel regression with bank fixed effects. The regression uses the ratio of commercial and industrial loans to total assets as the dependent variable. The independent variables are the interest and fee income on commercial and industrial loans relative to commercial and industrial loans and the logarithm of total assets. The parameters for the loan specific shocks are set using the mean and standard deviation of total charge offs relative to total assets for the 500 largest commercial banks.[35]

### 5.2. Yield Curve, Regulatory Constraints, and Loan Rates

In this section, we analyze the competing effects of the yield curve factors on the interest rate charged and quantity of loans offered by the bank operating under regulatory constraints. First, recall from (4) that the future yield curve factors include a percentage of the current shock to factors at time $t$, which persists until time $t + \tau$, and a random change, $Y$, in the yield curve factors from $t$ to $t + \tau$.[36] The effect of expected changes in the yield curve as well as the future random factor on the margin are given by

$$\frac{\partial E_t \left[ m_{\tau,t+\tau}^{j*} \right]}{\partial E_t [X_{t+\tau}]} = \left[ \frac{1}{2} d_1 - B_\tau \right] < 0, \text{ and } \frac{\partial m_{\tau,t+\tau}^{j*}}{\partial Y} = \left[ \frac{1}{2} d_1 - B_\tau \right] < 0, \tag{60}$$

since $B_\tau > d_1$. As a result, the loan margin falls when the expected and future levels of the yield curve increase. Notice, however, that the loan rate, $r^{j*}_{\tau,t+\tau} = m^{j*}_{\tau,t+\tau} + r_{\tau,t}(X)$, would increase and the quantity of loans would fall since the marginal cost of deposits, $d_1$, is higher.

Now, suppose the capital constraint is binding. Then, the effect on the interest rate margin of a change in the current yield is:

$$\frac{\partial E_t\left[m^{j\kappa}_{\tau,t+\tau}\right]}{\partial X} = -e^{-A^{\mathcal{P}}(\tau-t)}B_\tau - \frac{1}{\left(\gamma^j_{1,\tau} - \sigma_1\varepsilon^j_{\tau,t+\tau}\right)}\frac{\partial L^j_{\kappa,t+\tau}}{\partial X}. \tag{61}$$

The first term is the traditional effect of a higher expected future treasury rate $e^{-A^{\mathcal{P}}(\tau-t)}B_\tau$, which lowers the interest rate margin. The second term is the impact of the change in the yield on the quantity of loans.[37] In particular,

$$\frac{\partial L^j_{\kappa,t+\tau}}{\partial X} = -\frac{1}{\kappa_L}(1 - \kappa_T\xi)K^j_M(t)\mathcal{K}(\tau,X)p_K(t,X,\tau,Y)(\sigma_\mathcal{K}(\tau))^{-1}\left(X - \mu_\mathcal{K}(\tau)\right)$$

$$-\frac{1}{\kappa_L}\left[\frac{1}{\mathcal{M}(\tau,X)}\frac{\partial CCB}{\partial X} + c_b b_{3\tau}e^{-A^{\mathcal{P}}(\tau-t)}\left(\frac{P_{\tau,s}}{\bar{P}_{\tau,s}}\right)^+\right]$$

when $X > \mu_\mathcal{M}(\tau) > \mu_\mathcal{K}(\tau)$. \tag{62}

Equation (62) highlights two effects on the constrained level of loans: the first, is the impact of the investment decision of the trading desk and, the second, is the effect of CCB. Specifically, for an increase in the level of the yield curve, $X_1$:

1.  The first effect (see (34) when $X_1 > \mu_{\mathcal{K}1}(\tau)$), highlights the decrease in the trading desk's expected gross growth rate of capital which restricts lending. This effect is smaller for $X_1$ near $\mu_{\mathcal{K}1}(\tau)$, but increases for extreme levels of the yield curve, $X_1$.[38]
2.  The second term in (61) highlights the drop in CCB due to an increase in the current level of the yield curve (see (25) and for $X_1 > \mu_{\mathcal{M}1}(\tau)$). In this case, the bank is required to hold less capital and the constrained level of loans increases.

In short, the net effect of a higher level of the yield curve on the quantity of loans is dependent on the relative magnitudes of these two effects. Figure 6 highlights the impact of the two effects on the loan margin in (62) along with the traditional negative effect of a change in the current yield curve $-e^{-A^{\mathcal{P}}(\tau-t)}B_\tau$ (first effect in (61)). This latter effect is represented by the red line in the LHG of Figure 6. When the capital constraint binds, however, the first effect of a change in the current level of the yield curve on the trading desk investment decision is now represented by the blue line in the LHG of Figure 6. For the level of the yield curve close to the mean of the expected gross growth rate for the trading desk's capital, $\mu_{\mathcal{K}1}(\tau) = -0.0639$, the first effect in (62) is near zero. As a result, the blue and red curves in the LHG of Figure 6 intersect.[39] The second effect in (62) is from the CCB, and it is a smaller effect relative to the first. This results in a larger supply of loans, so that the blue curve in the LHG of Figure 6 would shift down by the magnitude of the drop in the CCB and the slope of the demand for loans.

The above analysis points to the importance of the change in the value of the trading desk's portfolio in determining the total effect of a change in the yield factor on the loan rate margin in (61). For example, when the losses to the trading desk portfolio due to an increase in yield factor are small, then the total effect on loan rate margin would be negative; the negative traditional impact of a change in the expected level of the yield curve is reinforced by the negative CCB effect in (62). In contrast, if losses to the trading desk portfolio are large the loan rate margin would increase. This happens for example when $X$ is larger than the stationary level of the yield curve $\bar{X}_1 = -0.0177$–the black dashed line in the LHG of Figure 6. These results are summarized in the following:

**Proposition 1.** *In general, the higher future expected yield curve factors have a positive effect on the constrained level of loans, (62), and a negative impact on the interest rate margin, (61), for $X > \mu_{\mathcal{M}}(\tau) > \mu_{\mathcal{K}}(\tau)$. However, an increase in the expected future yield curve factors could reduce lending and increase the interest rate margin, when the higher expected yield curve factors are far enough away from the mean $\mu_{\mathcal{M}}(\tau)$.*

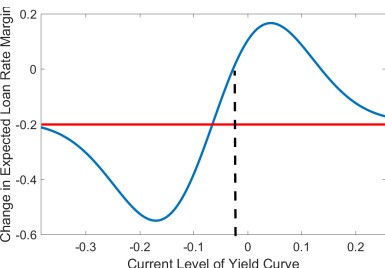 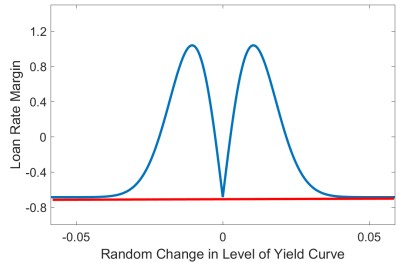

**Figure 6.** Loan Rate Margin (61) (LHG) and (64) (RHG).

We can also consider the impact of random changes in the future yield factors, $Y$, on bank lending and the loan margin–the RHG of Figure 6, when the capital constraint is binding. In this case, there is only one effect, that is, on the trading desk's capital, since the CCB is set based on expected yield curve factors rather than the random change in these factors. In the case of constrained loans (23), the trading desk's capital falls due to an increase in the absolute value of the random yield curve factors by (35). With less capital the bank must lend less:

$$\frac{\partial L^j_{\kappa,t+\tau}}{\partial |Y|} = -\frac{2}{\kappa_L}\left[(1 - \kappa_T\xi)K^j_M(t)\mathcal{K}(\tau, X)p_K(t, X, \tau, Y)(\sigma_K(\tau))^{-1}|Y|\right] < 0, \quad (63)$$

As a result, the impact on the loan rate margin is now given by:

$$\frac{\partial m^{j\kappa}_{\tau,t+\tau}}{\partial |Y|} = -B_\tau - \frac{1}{\left(\gamma^j_{1,\tau} - \sigma_1\varepsilon^j_{\tau,t+\tau}\right)}\frac{\partial L^j_{\kappa,t+\tau}}{\partial |Y|}. \quad (64)$$

If the random yield curve factors increase in absolute value, the probability distribution for the future capital of the trading desk falls, and the constrained loans decrease by the second term in (64). This leads to a larger loan margin under the capital constraint by (64) relative to the traditional effect, $B_\tau$, on the yield to maturity of treasury securities. This is represented in the RHG of Figure 6 by the shift from the red line to the blue curve.[40]

**Proposition 2.** *For shocks to the yield curve near zero, the random changes in the yield factors have a negative impact on the level of loans and the interest rate margin. For larger shocks to the yield curve, the interest rate margin can increase, when the capital loss by the trading desk dominates the traditional channel.*

Interestingly, the above Propositions 1 and 2 highlight the role of volatility of the yield factors in negatively affecting bank credit under the capital constraint. If the volatility is large enough, this will also result in higher interest rate margins. This raises the issue of the role of *forward guidance* by the central bank in mitigating the effect of volatility, by allowing banks to better anticipate future changes in yield factors. In particular, forward guidance would lessen the occurrence of extreme random values of the yield curve factors.

### 5.3. The Choice of Capital for the Loan Desk

In this section, we will derive the choice of capital for the loan desk by the COO. In order to do so, we need to evaluate the expected marginal value of capital (EMV) for the loan desk as in (43) or (44), where the Lagrange multipliers are replaced by (58) and (59).

$$
\begin{aligned}
E_t\left[p_M(2\tau, Y)\frac{\partial V}{\partial K_L^j(t+\tau)}\right] &= E_t\left\{p_M(2\tau, Y)\left[r^D(t+\tau)\tau\right.\right. \\
&\left.\left. + 2\chi Max\left[\frac{1}{\kappa_L}\left(r_{\tau,t+\tau}^{j\kappa} - r_{\tau,t+\tau}^{j*}\right)^+; \frac{1}{\alpha_\tau}\left(r_{\tau,t+\tau}^{jl} - r_{\tau,t+\tau}^{j*}\right)^+\right] + (\chi-1)\tau\right]\right\}.
\end{aligned}
\tag{65}
$$

Note, henceforth, we make use of the option terminology to highlight better the possibility that the regulatory constraints may or may not be binding, which, as we will see shortly, could be due to the actions of the trading desk or due to monetary and financial market shocks. This, in turn, will impact the EMV and choice of capital. In this sense, capital becomes more valuable to the bank as constraints are more likely to be binding. Thus, $(\bullet)^+$ is zero when either interest rate spread is negative. $Max[x; y]$ means that the COO calculates the option payoff under the liquidity and capital constraints separately and chooses the largest option payoff.[41]

The EMV can then be viewed as an option with expiration date $t+\tau$ and strike price given by $r_{\tau,t+\tau}^{j\kappa}$ or $r_{\tau,t+\tau}^{jl}$. The payoff of the option under the capital constraint is obtained by substituting the constrained (56) and unconstrained (55) loan rate margin into (65). In addition, replacing the level of loans under the capital constraint with (23) and (18), we get

$$
\begin{aligned}
\frac{1}{\kappa_L}\left(r_{\tau,t+\tau}^{j\kappa} - r_{\tau,t+\tau}^{j*}\right)^+ p_M(2\tau, Y) &= \frac{1}{\left(\gamma_{1,\tau}^j - \sigma_1\varepsilon_{\tau,t+\tau}^j\right)}\left(\frac{c_b}{\kappa_L}\left(\frac{P_{\tau,s}}{\bar{P}_{\tau,s}} - 1\right)^+ + \frac{1}{2}\left(\gamma_{0,\tau}^j + \sigma_0\varepsilon_{\tau,t}^j\right)\right. \\
&\quad - \frac{1}{\kappa_L}K_L^j(t+\tau) - \frac{1}{2}\left(\gamma_{1,\tau}^j - \sigma_1\varepsilon_{\tau,t}^j\right)\left(c^j + d_0 + d_1 X(t+\tau)\right) \\
&\quad \left. - \frac{1}{\kappa_L}(1-\kappa_T\xi)K_M^j(t+\tau) + \left(L_{2\tau,t+\tau}^j + L_{2\tau,t}^j\right)\right)^+ p_M(2\tau, Y).^{42}
\end{aligned}
\tag{66}
$$

As alluded to earlier, Equation (66) demonstrates that the probability the capital constraint is binding depends on the CCB, future values of the yield factors, market valuation as reflected in the SDF, and on the actions of the trading desk. In particular, this payoff is dependent on the probability distribution for the random change in the factors (5), the stochastic discount factor (10) and the random changes in the gross growth rate of capital for the trading desk (36). Each of these distributions are normal with mean zero and distinct variance-covariance matrices. Thus, the valuation of these options is more complicated than in the Black-Scholes case, since both the strike price and the unconstrained interest rate on loans are random and this randomness has uncertainty arising from the level, slope and curvature of the yield curve.

In Figure 7, we plot the payoff for the EMV for the loan desk (66). In the LHG, we keep the capital for the trading desk constant and the current level of the yield curve at its stationary value $\bar{X}_1 = -0.0177$. For a given capital of the loan desk, say $K_L^j(t) = 0.1$, then the EMV is at its lowest, or the option payoff is at its lowest, at the mean of zero for the future level of the yield curve $Y_\tau$. This is because at this level of the $Y$, the trading desk's hedging portfolio is likely to result in higher capital for the bank. We observe that the hedging behavior by the trading desk confers expected benefits to the lending operation and the overall bank, by providing more capital. This, in turn, lowers the probability of the capital constraint binding. Recall that the random component of the gross rate of return on the capital of the trading desk is normally distributed following (35). As a result, the payoff on the option can be represented as an inverted normal distribution of Figure 7. In the RHG, we limit the future level of the yield curve to be positive, so that we can observe

how the payoff varies with $K_M^j(t)$ from 0.05 through 0.2. In this case, the payoff for the EMV for the loan desk is negatively influenced by an increase in the capital of the trading desk. This impact is largest when the future level of the yield curve is near the mean of zero. Again, with $Y$ at its mean of zero, the hedge of the trading desk delivers higher capital for the trading desk, and in turn for the loan desk and for the bank.

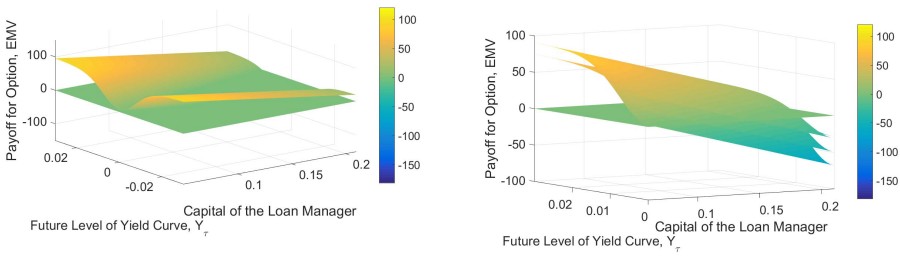

**Figure 7.** Payoff for the Embedded Option (66).

### 5.4. The Probability of Distress and Loan Desk's Capital

In order to calculate the EMV in (65), we first need to evaluate the embedded option. We examine the payoff (66) for this option in Figure 7 for values of capital of the trading desk $K_M^j = 0.05$ through $K_M^j = 0.2$. If the trading and loan desks' capital are given, then the future level of the yield curve determines the payoff of the option. We plot this figure for positive future values of the yield curve.[43] Note, the option is in–the–money, or the EMV is at its highest, when the level of the yield curve is at its extreme values. However, when the future yield curve is near its zero mean the option is out–of–the–money, or the EMV is at its lowest. As a result, the EMV for the loan desk behaves like a combination of long European put and call options. The options expire at the next period. Note that the strike price for the put option is below that for the call option. The combined options are referred to as a *Long Strangle*, an option that is in–the–money only for extreme values of the level of the yield curve.

For extreme future values of the yield curve $Y$, be it negative or positive, the hedging strategy by the trading desk manager does not pay off, and the portfolio investment results in capital losses for the trading desk. This, in turn, reduces the capital for the loan desk and for the overall bank. This also implies that high volatility of future yield curve factors is likely to increase the chance of capital losses for the trading desk and, in turn, exacerbate capital losses for the rest of the bank. This raises the probability that the capital constraint binds and increases the EMV. On the other hand, for normal conditions, that is, when future rates $Y$ are around their mean, the hedging behavior by the trading desk *insures* the loan desk and the bank against interest rate risk, and, thereby, lowers the chance that the capital constraint binds, which reduces the EMV.

Next, we need to identify when the payoff for the Long Strangle option in Figure 7 and (66) is zero.[44] For given values of variables $X, K_M^j, K_L^j(t+\tau), \varepsilon_{\tau,t}^j$, we use the notation $\rho_\kappa$ for the critical value for the future level of the yield curve at $t+\tau$ such that $r_{\tau,t+\tau}^{j\kappa} = r_{\tau,t+\tau}^{j*}$, and the payoff of the option crosses the zero plane.

$$\rho_\kappa = \rho_\kappa\left(\tau, X, K_M^j, K_L^j(t+\tau), \varepsilon_{\tau,t}^j\right). \tag{67}$$

This critical value of the level of the yield curve is increasing in the capital of the loan desk and trading desk–since the EMV is decreasing–but is decreasing in the random change in the demand for loans, $\varepsilon_{\tau,t}^j$.[45] The impact of the interest rate factors is determined by the following

**Condition**:

1.  $\left(\gamma_{1,\tau}^j - \sigma_1\varepsilon_{\tau,t}^j\right)d_1 > 0$, and $X > \mu_\mathcal{K}$ is close to $\mu_\mathcal{K}$

2. $\left(\gamma_{1,\tau}^{j} - \sigma_1 \varepsilon_{\tau,t}^{j}\right) d_1 > 0$ is close to zero and $X > \mu_{\mathcal{K}}$.

A higher level of the yield curve always leads to a higher deposit rate. This higher deposit rate results in a higher unconstrained interest rate margin by (55). $X - \mu_{\mathcal{K}}$ determines the slope of the gross growth rate of capital.

Under Condition 1 the future critical factor is increasing in the interest rate factors, while it is decreasing under Condition 2.

In the LHG of Figure 8, we keep the level of the yield curve at its stationary value and graph the cutoff versus the capital of the loan desk. This graph represents a tranche of Figure 7. It is constructed using all the planes that go through zero for different values of capital for the loan desk $K_L^j$. These critical values are portrayed in Figure 8. Consider the LHG in Figure 8. For given values of $X_1$ and $K_M^j$, the strangle (66) is in–the–money outside any of the curves. In this case, the EMV is increasing since the capital constraint on loans (23) is binding. This is likely to happen for extreme values of the random shocks $Y$. In this case, the values of the capital stock for the trading desk, $K_M^j(t + \tau)$, and the stochastic discount factor, $\frac{M_{2\tau,t}}{M_{t,t}}$, in (66) are lower.[46] Inside the curve the capital constraint does not bind and the payoff of the strangle is zero.[47]

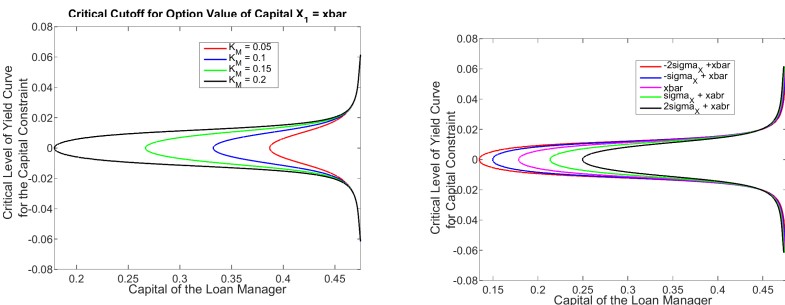

**Figure 8.** Critical Cut off $\rho_\kappa$, (67), such that Capital Constraint Binds, (23).

The LHG in Figure 8 shows that for higher values of capital for the trading desk, the capital needed by the loan desk to maintain a positive critical cut off is lower. That is, successful hedging by the trading desk results in higher capital overall, as it buffers the loan desk against interest rate risk. Outside and to the left of the curves in the LHG the capital constraint binds and the payoff of the strangle is positive. Indeed, when the magnitude of future yield rates $Y$ is large, the trading desk's hedging strategy does not shield the trading and lending operations as well as the bank against interest rate risk, and, with losses the capital constraint will likely bind.

The RHG in Figure 8 highlights the impact of a change in the level of the yield curve, given $K_M^j = 0.2$. A decrease in the absolute level of the yield curve will cause the critical value to increase for given capital for the loan desk and $\rho_\kappa \geq 0$. In this case, the capital constraint is less likely to be binding, i.e., the region inside the curve in which the constraint does not bind has increased. This conclusion follows from the capital gain on the trading desk's portfolio as the level of the yield curve declines, as shown in the bottom graph in Figure 5.

Given the critical cutoff (67), we can now calculate the cumulative probability that the bank becomes capital constrained, or in distress:

$$Prob\left[\left(r_{\tau,t+\tau}^{j\kappa} - r_{\tau,t+\tau}^{j*}\right) \geq 0\right] = 2\Phi\left[\rho_\kappa\left(\tau, X, K_M^j(t), K_L^j(t+\tau), \varepsilon_{\tau,t}^j\right)\right] \text{ where } \rho_\kappa > 0. \quad (68)$$

Consequently, we can see the impact of changes in the interest rate factors, the trading desk's capital and the loan manager's capital on the likelihood that the bank becomes distressed. In particular, a lower $\rho_\kappa$ leads to a higher chance that the bank will be in distress and, as a result, be subject to restrictions by the financial regulator. For given capital of the

loan desk, a lower capital for the trading desk, $K_M^j$, leads to a lower critical value, $\rho_\kappa$, in the LHG of Figure 8, which means the overall bank has a larger possibility of being capital constrained. We can now summarize our discussion in the following result:

**Proposition 3.** *The probability of a bank becoming distressed, given by (68), is decreasing in the capital of the trading desk and the loan desk, while it is increasing in the shock to the demand for loans. This probability of distress is decreasing in the yield curve factors under Condition* 1 *and increasing under Condition* 2.

The above analysis allows us to flesh out the various components of EMV for the loan desk. In particular, it is a combination of Black–Scholes formulas at the two critical values for $Z$, i.e., $\rho_l$ and $\rho_\kappa$. When the capital constraint binds, i.e., $\lambda_1 > 0$ and $\lambda_2 = 0$, the EMV for the loan desk can be written as[48]

$$
\begin{aligned}
EMV(X, K_M^j, K_L^j(t+\tau)) &\equiv \mathcal{M}(2\tau, X) E_t\left[p_M(2\tau, X)\frac{\partial V}{\partial K_L^j(t+\tau)}\right]\\
&= \mathcal{M}(2\tau, X)\Bigg\{(d_0 + d_1\mu(\tau, X) + 1)\tau + (\chi - 1)\tau\\
&+ \sum_{i=1}^{S} Pr\left(\varepsilon_{\tau,t}^j = \varepsilon_i^j\right)\frac{2\chi}{\kappa_L\left(\gamma_{1,\tau}^j - \sigma_1\varepsilon_i^j\right)}\left[\frac{CCB(X)}{\mathcal{M}(2\tau, X)} + \left\{\frac{1}{2}\left(\gamma_{0,\tau}^j + \sigma_0\varepsilon_{\tau,t}^j\right)\right.\right.\\
&- \frac{1}{\kappa_L}K_L^j(t+\tau) - \frac{1}{2}\left(\gamma_{1,\tau}^j - \sigma_1\varepsilon_{\tau,t}^j\right)\left(c^j + d_0 + d_1\mu(\tau, X)\right)\\
&- \frac{1}{2}\left(\gamma_{1,\tau}^j - \sigma_1\varepsilon_{\tau,t}^j\right)d_1\frac{\sqrt{det(\sigma_M(\tau))}}{\sqrt{det(\sigma_M(\tau) + \sigma_Y(\tau))}}\Phi\left(\Sigma_Y(\tau)^{-1}\rho_\kappa\right)\\
&- \frac{(1 - \kappa_T\xi)}{\kappa_L}K_M^j(t)\mathcal{K}(\tau, X)\frac{\sqrt{det(\sigma_M(\tau))}}{\sqrt{det(\sigma_M(\tau) + \sigma_K(\tau))}}\Phi\left(\Sigma_K(\tau)^{-1}\rho_\kappa\right)\Bigg\}2\Phi\left(\Sigma_M^{-1}\rho_\kappa\right)\Bigg]\Bigg\}.
\end{aligned}
\tag{69}
$$

Given this explicit formula for the EMV for the loan desk, which can be understood as a combination of embedded options, we can calculate the option delta $\Delta_\kappa = \frac{\partial EMV(X, K_M^j)}{\partial K_L^j(t+\tau)} < 0$, which satisfies the second order condition to issue equity or pay dividends, $\Delta_\kappa < 0$.[49]

**Proposition 4.** *The COO's choice of capital for the loan desk is given by*

$$
K_L^{j*}(t+\tau) = K_L^j(\tau, K_M^j, X).
\tag{70}
$$

Figure 9 highlights the choice of the capital for the loan desk for a given level of the trading desk's capital and interest rate factor. Note that EMV for the loan desk is decreasing in the capital for the loan desk.

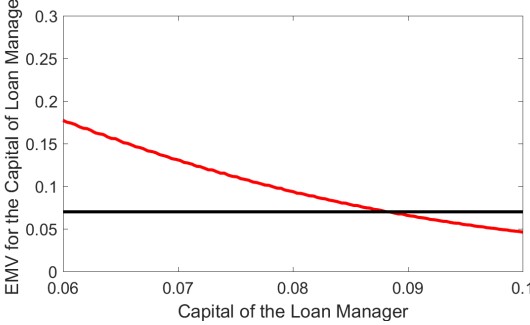

**Figure 9.** EMV of $K_L^j$ (69) and optimal $K_L^j$ (70).



## 6. Conclusions

This BHC model can be used to understand important issues in the management and regulations of these institutions. First, is there a benefit from having both lines of business, or should the trading desk be separated from the rest of the bank? We believe this is an important question, especially in light of the impact of the recent bank regulations on proprietary trading business at banks–with a number of these banks shutting off their proprietary trading businesses. Using the BHC model, Chami et al. (2017) in their section VI shows that high leverage by the trading desk is harmful to the overall bank, and the COO scales back the portfolio of the trading desk. In fact, there exists a leverage level such that the second order condition fails, and the trading desk is ring-fenced. Thus, the overleveraged position can be decreased by the bank regulator by increasing the regulatory capital weights for treasury securities, i.e., $\kappa_T$ and $\alpha_T$.

Second, having established the relation between yield curve factors and financial markets, regulations, and the choices of trading and loan desks, Chami et al. (2017) in their Section VII analyze the impact of monetary policy on the optimal level of bank capital. They argue that the impact of a change in monetary policy on bank profitability and distress is dependent on the specific bank circumstances such as its elasticity of loan demand and the level of the yield curve. It is also clear that monetary policy stance has financial stability implications through its impact on bank distress. Some of the channels described above are likely to amplify the negative externality, with trading desk behavior, for example, amplifying bank distress, while others are likely to mitigate the impact, such as the CCB. Irrespective of the conflicting effects described above, our framework highlights the close link of monetary policy to financial stability, and provides insight on the need to focus on bank specific reaction to each of the above effects.

A number of important implications and recommendations for BHC regulatory policies can be gleaned from this work. First, a less risk averse manager chooses a riskier portfolio which results in higher expected losses in capital for the trading desk. This translates to lower capital for lending, and causes the regulatory constraints to likely bind and leads to fewer loans. Consequently, bank management should specify to the trading desk manager the level of risk aversion and leverage that is acceptable. Second, hedging by the trading desk manager does not necessarily immunize the bank against interest rate risk. Extreme future values of the yield curve factors or high volatility of these factors could subject the trading desk to losses, which, in turn, would reduce the capital cushion provided to the lending operation and lead to higher possibility of bank distress. Thus, there is a need to distinguish better between rogue trading, which has received attention in the public domain and in policy circles, and hedging gone bad due to uncertainty regarding future rates. One solution to this is better communication by the central bank about its future policy rate setting intentions. This forward guidance by the central bank would help banks better position their portfolios, and limit the losses from the underlying interest rate volatility. Finally, the concern for bank distress and its implications for the larger economy have galvanized policymakers and bank regulators to enact rules and laws that require the separation of proprietary trading from the lending business. However, there are benefits from maintaining both businesses, as the trading desk confers insurance benefits on the rest of the bank by reducing interest rate risk and lowering the probability of bank distress. This is true despite the fact that both business lines are subject to the same market risk. However, as discussed earlier, we also provide conditions where overleverage and aggressive risk behavior on the part of the trading desk manager result in bank distress. In this case, the COO may find it optimal to ring-fence the trading business. The policy question then is whether it is possible to keep the benefit from maintaining the two businesses while ensuring, to the extent possible, that the trading business does not unnecessarily impose negative externality on the rest of the bank?

**Supplementary Materials:** The following are available online at https://www.mdpi.com/article/10.3390/jrfm15050206/s1, (See Arnold (1974); Basel Committee on Banking Supervision (2011); Begley et al. (2015); Calin et al. (2011); Evans (2002); Hijab (1987); Petersen and Pedersen (2008); Strauss (2008) and Strichartz (2008)). Supplementary of What's Different about Bank Holding Companies?

**Author Contributions:** R.C., T.F.C., J.M. and C.R. provided contributions to all parts of the paper. All authors have read and agreed to the published version of the manuscript.

**Funding:** Cosimano was funding as a Visiting Scholar by the Institute for Capacity Development in the International Monetray Fund.

**Institutional Review Board Statement:** The views expressed in this paper are those of the aurhors and do not necessarily represent the views of the IMF, its Execute Board, or IMF management.

**Informed Consent Statement:** No human or animal subjects were used in the conduct of this research.

**Data Availability Statement:** All the data used in this paper are publicly available and described in the endnotes.

**Conflicts of Interest:** All the authors have been funded by Academic Institutions and the IMF, which do not impose any conflict of interest.

## Notes

[1] Most US BHCs have now spun off their trading desks while retaining trading in government securities, munis, as well as trading on behalf of their customers-all allowed under the Volker rule.

[2] De Nicolò et al. (2014) also develop a dynamic model of banking to analyze aspects of Basel III, however they do not consider how the hedging strategy of the trading desk can generate insurance for the loan desk.

[3] Avaham et al. (2012) provide the rules which distinguish between BHC and FHC within the US. A large percentage of commercial banks are part of either a BHC or FHC. The BHC may be limited in their ability to trade marketable securities, but FHC are not as restrictive. Universal Banks are similar to FHC in that they provide a large menu of financial services—See (Morrison 2012). The distinguishing element of our bank is the presence of both trading and loan desks. We refer to these institutions as BHCs, for simplicity.

[4] Source Flow of Funds Z1 9 June 2016 Table L.108 and Top Tier BHC as of 30 June 2015 from Federal Reserve Board and Bank of Chicago, respectively.

[5] Froot and Stein (1998) consider the impact of the financial markets through the CAPM model. Our framework is developed in the context of no arbitrage term structure models with optimal behavior by the bank that is subject to Basel III. He and Krishnamurthy (2013) provides a model with a specialist desk similar to a trading desk that is subject to capital constraints. Nagel and Purnanandam (2016) consider a bank with overlapping loans and no trading desk in which the underlying projects, financed by these loans, follow a stochastic process. They then value the option value of loans, debt and equity.

[6] The model of the BHC industry follows Chami and Cosimano (2010). Each BHC operates with some monopoly power, but there is always the threat of price competition. For simplicity we only look at the optimal behavior of the BHC under some monopoly power. Chami and Cosimano (2010) discuss the evidence in favor of monopoly power in banking. See Chami and Cosimano (2010); Schliephake and Kirstein (2013); and Corbae and D'Erasmo (2014) for further discussion of oligopoly and capital requirements.

[7] We use a continuous time version of the term structure model of Joslin et al. (2011). This allows us to derive probability distributions for key endogenous variables using Forward Kolmogorov equations following Karatzas and Shreve (1988).

[8] The trading desk problem is an extension of Liu (2007) and Sangvinatsos and Wachter (2005).

[9] This variance is the solution to a Ricatti differential equation. The solution is found by using recursive rules, which are implemented in the lyap subroutine in Matlab with inputs $A^{\mathcal{P}}$ and $\Sigma_X$.

[10] See Cosimano and Ma (2018) Section 2 for the proof.

[11] See Equation (25) in Cosimano and Ma (2018). Recall that the unanticipated shock to the interest rate factor is log-normally distributed. Taking the conditional expectation converts the shock into a time-dependent term only, which we include in $\mathcal{M}(\tau)$ to simplify the notation.

[12] The transitional probability (10) is found by solving the Forward Kolmogorov equation in Cosimano and Ma (2018).

[13] We use the data in van Dijk et al. (2014) which they generously provide on the journal's website.

[14] Specifically, we subtract the mean and divide by the standard deviation from each series.

[15] However, it is important to note that in this affine term structure model and given the negative eigenvalues of $A^{\mathcal{Q}}$ there is no exact mapping between these empirically defined factors and the extracted latent factors, as in Nelson-Siegel model (see Diebold and Li (2006)). The Section S6.3 demonstrates that the factors and latent variables capture the monetary policy for the US 1990–2013.

See Bauer et al. (2012, 2014); Joslin et al. (2013, 2014); and Gürkaynak and Wright (2012) for work on the relation between the macro economy and the term structure factors. We show the relation between the yield curve factors and the Taylor rule for US monetary policy. The level factor places the strongest weight on monetary policy variables. The second and third latent factors are better explained by the US monetary policy. Therefore, this comparison is meant to show that the latent factors from our affine term structure model approximate fairly well these empirically defined level, slope and curvature factors.

[16]  In principle the COO could also choose the leverage ratio to control the marginal product of capital for the trading desk. However, the COO has an instrument, capital for the trading desk, to control this marginal product.

[17]  The date of issue for these securities is not relevant, since the trading desk can always swap old securities with new securities with the same maturity. In practice, the 2 period security can be replaced with a $j$ period security and the 3 period security can be replaced with the $k$ period security without any change in the analysis.

[18]  If a secondary market is not present in the country, then both the treasury and loan decisions would be made in discrete time.

[19]  This short term funding can be in the form of repurchase agreements on the marketable securities. If the bank is not able to borrow in the markets, the bank would then have to borrow from the central bank at some interest rate.

[20]  See pages 11–12 of the Supplementary for the derivation.

[21]  In the Section S1, it is shown that the coefficients $\sigma_J(\tau)$, $\mu_J(\tau)$ and $J(\tau)$ satisfy three ordinary differential equations. These coefficients are conformable to $X$.

[22]  In the Section S2, we provide the solution procedure which is similar to that for the stochastic discount factor.

[23]  This procedure is the same as for Equation (10).

[24]  The details for this derivation is contained in the Sections S1 and S2.

[25]  The impact of leverage is discussed on pages 24–25 of Chami et al. (2017).

[26]  To simplify notation we use $p_M(2\tau, Y)$ for the transition probability $p_M(t, X, 2\tau, Y)$ (10), since it is not dependent on $t$, $X$ in this case. The value function is dependent on the capital of the trading and loan desk, when they make their decisions. However, the value of the bank after the COO makes her decisions is dependent only on the yield curve factors, since her decisions are dependent on the optimal decisions of the two desks.

[27]  See Bessembinder and Zhang (2013) for a survey of this work.

[28]  The second order condition for payment of dividends or issuing stock is true, as long as the expected marginal value of capital is decreasing in the capital for the loan desk.

[29]  See for example, Barnea and Kim (2014); and Corbae and D'Erasmo (2014).

[30]  The solution for the bank's problem is found for an arbitrary number of states while the discussion is in terms of just two.

[31]  The data from Report of Condition and Income data for Commercial Banks was organized for the 500 largest commercial banks by Sebastian Rolands.

[32]  Note that here the regressor X(t) takes the same value across different banks at any given time point, and thus this eliminates the concern of endogeneity in the cross-sectional variations. Furthermore, since our goal is to document the average quantitative response of these rates paid to deposits and reserve balances to the level of the yield curve at the national level, we do not attempt to trace this response to any type of structural shock nor do we attempt to establish any causality. Therefore, as long as the level of the yield curve can summarize well enough the overall macroeconomy and policy expectations, this simple linear regression would yield valid quantitative responses that are needed in our simulation exercises. In addition, the simulations were checked for changes in these coefficients by $\pm$ one standard deviation of the coefficient on the level. We found that the qualitative results reported here were robust to these changes. The details of the data and regressions are reported in Section S6.2 of the Supplementary.

[33]  See Chami and Cosimano (2010) for example. Inclusion of a model of deposit insurance as in De Nicolò et al. (2014) would lower the cost of deposits for the bank. This would be reflected in a reduction of $d_0$ and/or $d_1$. In our robustness checks, we found that changing these parameters did not materially impact the conclusions of this paper.

[34]  The properties of the two period loans are shown in the Supplementary Section S3 to be proportional to the one period loans and dependent on the expected stochastic discount factor, $\mathcal{M}(\tau, X)$, in Equation (9) over the additional period of the loan.

[35]  These parameters are determined in the Supplementary Section S6.2 using a binomial probability distribution. We set the probability of the bad outcome being $p = 0.00566$ which corresponds to the average charge offs across all banks in the panel.

[36]  Recall $E_t[X_{t+\tau}] = e^{-A^P(\tau-t)}(X - \bar{X})$.

[37]  Note that when the capital constraint binds the marginal cost of deposits, $d_1$, is no longer relevant, since the quantity of loans associated with the capital constraint (23) precludes equating marginal revenue with marginal cost. As a result, the loan rate margin rises and fewer loans are issued.

[38]  The subscript 1 refers to the first (level) factor for the yield curve.

[39]  This simulation is for the case of $\gamma^j = 10$, $\xi = 1$, and $K_M^j = 0.05$.

[40]  Note, this effect is stronger, as long as, its magnitude is below a one standard deviation shock to the level of the yield curve $\sigma_K(\tau) = 0.0104$. This follows from the inflexion point of the Gaussian probability density function.

41    In the Supplementary Section S4 we calculate the option value of the capital constraint and liquidity constraint separately. This determines the two elements in the maximization in (65), which the COO compares to find the option value of the straddle.

42    For the payoff of the option under the liquidity constraint, replace the level of loans under the liquidity constraint with (22) and (18). See Supplementary Section S4.1 for the derivation under the capital constraint and Supplementary Section S4.2 for the liquidity constraint.

43    There is a mirror image of this curve for negative values of the future yield curve.

44    The zero payoff also occurs at the negative value of the cutoff in Figure 7.

45    The explicit formula for solving for this cutoff is in the Supplementary Section S4.

46    For valuing the option, we use the larger critical point for the cutoff (67), so that the probability the capital constraint binds is $2\Phi\left(\rho_\kappa\left(\tau, X, K_M^j(t), K_L^j(t+\tau), \varepsilon_{\tau,t}^j\right)\right)$ by (26), because the payoff of the option is symmetric about zero in Figure 7.

47    To the left of any given curve the probability the capital constraint binds is one.

48    We also have to account for the random shock to the demand for loans $\varepsilon_{\tau,t}^j$. It is assumed to have a discrete distribution with $S$ values $\varepsilon_i^j$. Here, it is assumed for simplicity that the demand for loans is independent of the interest rate factors. To account for a correlation among the interest rate factors and the demand for loans, one can assume: The demand for loans has a normal distribution, which is correlated with the normal interest rate factors. One then solves the option problem conditional on the shock to the demand for loans followed by a discrete representation of the probability distribution for the loan demand shock using a Gauss–Hermite polynomial approximation. In the Supplementary Section S4.2 we also provide the formula when the liquidity constraint binds.

49    In the Supplementary, Section S5 we derive the needed conditions for an interior solution for the issuing of equity or payment of dividends, which is stated in (43) or (44).

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
