# Peer review of "What’s Different about Bank Holding Companies?"

_jrfm, doi:10.3390/jrfm15050206_

Round 1
Reviewer 1 Report
I believe that this paper makes a significant contribution to the literature and deserves publication.
Author Response
Thank you for your review. The paper has been cut from 57 to 46 pages following your recommendation. This reduction has been accomplished by
1.) Removing impulse response functions on page 10.
2.) Deleted figure 4.
3.) Deleted section IV B pp. 25-26 on leverage.
4.) Summarized Sections VI, VII and VIII in a conclusion on page 41-42 of the new manuscript.
5.) Deleted unnessary references.
Sincerely yours,
Thomas Cosimano
Reviewer 2 Report
The paper is too long, more than 50 pages for the main text and over 100 in total
The model gives a significant attention to the risk attitude of the trading activity, while the lending activity risk levels and targets is not given the same attention.
Results are mainly justified by a theoretical model, which is well developed and linked to the previous literature, but very complex to follow.
The results are in line with the previous knowledge, confirming that the trading activity, when properly targeted, can mitigate the lending activity risks, but when the risk attitude is too high, it can higher the bank default risk.
My suggestion for the Authors is to shorten the paper, and to concentrate the attention to the content and significance of each model and results.
Author Response

(The authors gave the same response as above.)

Reviewer 3 Report
Thank you for the chance to review this paper. I usually dont get to review papers in High Economics - with full blown modelling done via mathematical proofs done well. This is clearly a paper which could appear in a top 10 journal - with a bit of pruning. It was/is an IMF working paper (albeit I got a version with the appendices).
The version I received is of course technically competent, interesting and innovative. Few try to open the black box I-bank desks. Of course, I could make observations about particular points. But the paper has clearly been well-shopped.
The only issue I might raise is the journal's word/page limit. I am guessing you are coming to the journals to get a fuller airing of your work, rather than the limited one an IMF working paper requires. The appendix goes only online - but the body is still a good 50 pages, with 1.25-ish line spacing.
Would you be interested in condensing it? And versioning? Or chopping it into a theory and more applied paper? If you are just trying to get this out with as little effort as possible, this might be a quick and easy approach.
I assume you are not submitting here to max-out your journal ranking scores. Otherwise, you would have chosen the QJE, JBF, JoF, etc. So the main value I could possible offer is to suggest ways of targeting the piece to the mixed audience that the JRFM mostly engages with. If you would like to do that (target the paper toward a non-theoretical, non-PhD audience), I am happy to provide suggestions to get to that end. And maybe chop it into 3-ish papers pitched at the relevant audiences. So you can get the right parts of your paper out to the right audiences. But I dont want to seem over-eager.
So I recommend acceptance without revisions. That way, you have the option to just push it through (if word constraints dont bind). And if you decide to take up my offer to target the piece, you might get 2-3 papers that target specific groups of readers, in a way that just reprinting the IMF Working Paper can not do. More real coverage and impact.
This paper was really written for me and my ilk. I enjoyed it very much and wont soon forget it. Thanks for giving me the chance to see it!
Author Response

(The authors gave the same response as above.)
